# Explicit solution of divide-and-conquer dividing by a half recurrences with polynomial independent term

**Tomás M. Coronado**[1,2☯], **Arnau Mir**[1,2☯], **Francesc Rosselló**[1,2☯] *

**1** Dept. of Mathematics and Computer Science, University of the Balearic Islands, Palma, Spain, **2** Balearic Islands Health Research Institute (IdISBa), Palma, Spain

☯ These authors contributed equally to this work.
* cesc.rossello@uib.eu

**Data Availability Statement:** The relevant data are fully available in Github at https://github.com/biocom-uib/divide_and_conquer.

**Funding:** All authors were partially funded by the Spanish Ministry of Science and Innovation and the

## Abstract

Divide-and-conquer dividing by a half recurrences, of the form

$$x_n = a \cdot x_{\lceil n/2 \rceil} + a \cdot x_{\lfloor n/2 \rfloor} + p(n), \quad n \geqslant 2,$$

appear in many areas of applied mathematics, from the analysis of algorithms to the optimization of phylogenetic balance indices. These equations are usually "solved" by means of a Master Theorem that provides a bound for the growing order of $x_n$, but not the solution's explicit expression. In this paper we give a finite explicit expression for this solution, in terms of the binary decomposition of $n$, when the independent term $p(n)$ is a polynomial in $\lceil n/2 \rceil$ and $\lfloor n/2 \rfloor$. As an application, we obtain explicit formulas for several sequences of interest in phylogenetics, combinatorics, and computer science, for which no such formulas were known so far: for instance, for the Total Cophenetic index and the rooted Quartet index of the maximally balanced bifurcating phylogenetic trees with $n$ leaves, and the sum of the bitwise AND operator applied to pairs of complementary numbers up to $n$.

## Introduction

*Divide-and-conquer dividing by a half* recurrences, of the form

$$x_n = a \cdot x_{\lceil n/2 \rceil} + a \cdot x_{\lfloor n/2 \rfloor} + p(n), \quad n \geqslant 2, \tag{1}$$

appear in many areas of applied mathematics. As evidence, for instance, the *On-line Encyclopedia of Integer Sequences* (OEIS, https://oeis.org) contains more than 200 integer sequences with the keyword "divide and conquer', many of which satisfy a recurrence like (1) [1].

The most popular and best studied setting where such recurrences arise is in Computer Science, and more specifically in the analysis of balanced divide and conquer algorithms. This type of algorithms solve a problem by splitting its input into two or more parts of the same size, solving (recursively) the problem on these parts, and finally combining these solutions into a solution for the global instance [2, §2.6–7]. Typical examples of this strategy are the heapsort and mergesort algorithms and several fast integer and matrix multiplication methods.

European Regional Development Fund through projects PGC2018-096956-B-C43 and PID2021-126114NB-C44 (FEDER/MICINN/AEI). The funders had no role in study design, data collection and analysis, decision to publish, or preparation of the manuscript.

**Competing interests:** The authors have declared that no competing interests exist.

When the input is split into two parts of the same size (up to a unit of difference, when the size of the original input is odd) and both parts contribute equally to the final solution, the algorithm's running time $t_n$ on an instance of size $n$ satisfies a recurrence of the form

$$t_n = a \cdot t_{\lceil n/2 \rceil} + a \cdot t_{\lfloor n/2 \rfloor} + p(n)$$

where the independent term $p(n)$, called in this context the *toll function*, represents the cost of combining the solutions of the subproblems into a solution for the original problem.

Our interest in this type of recurrences stems from the study of phylogenetic balance indices. A *phylogenetic tree*, the standard representation of the joint evolutionary history of a group of extant species (or other Operational Phylogenetic Units, OPU, like genes, languages, or myths), is, from the formal point of view, a leaf-labeled rooted tree. In a phylogenetic tree, its leaves represent the species under study, its internal nodes represent their common ancestors, the root represents the most recent common ancestor of all of them, and the arcs represent direct descendance through mutations [3, 4]. A phylogenetic tree is *bifurcating*, or *fully resolved*, when every internal node has two direct descendants, or *children*.

Biologists use the *shape* of phylogenetic trees, that is, their raw branching structure, to deduce information on the forces beneath the speciation and extinction processes that have taken place [5]. A popular set of tools used in the analysis of phylogenetic tree shapes are the *shape indices* [6], and among them the *balance indices*, which measure the propensity of the direct descendants of any given node in a tree to have the same number of descendant leaves. Many balance indices have been proposed in the literature: see, for instance, [3, 7–15] and the recent survey [16] and the references therein. Let us recall here the four indices that appear below in the Examples section:

- The *Sackin index* $S(T)$ of a tree $T$ [14, 15] is the sum of its leaves' *depths* (that is, of the lengths of the paths from the root to the leaves).

- The *Colless index* $C(T)$ of a bifurcating tree $T$ [7] is the sum, over all its internal nodes $v$, of the absolute value of the difference between the numbers of descendant leaves of the pair of children of $v$.

- The *Total Cophenetic index* $\Phi(T)$ of a tree $T$ [12] is the sum, over all its pairs of internal nodes, of the depth of their lowest common ancestor.

- The *rooted Quartet index* $rQI(T)$ of a bifurcating tree $T$ [17] is the number of its subtrees of 4 leaves that are fully symmetric.

Any sensible balance index should classify, for each number of leaves $n$, as most unbalanced trees the *rooted caterpillars* with $n$ leaves $K_n$ —the bifurcating trees such that all their internal nodes have a leaf child— and as most balanced bifurcating trees the *maximally balanced trees* with $n$ leaves $B_n$ —where, for each internal node, the numbers of descendant leaves of its pair of children differ at most in 1— (cf. Fig 1). The four indices described above satisfy this condition [16].

Now, given a balance index $I$, it is unreasonable to expect it to allow for the direct comparison of the balance of trees with different numbers of leaves. A possible way to circumvent this difficulty is to normalize it to [0, 1] for every number of leaves $n$ by subtracting its minimum value for this number of leaves and dividing by its range of values on the space of trees with $n$ leaves [15, 18]. To perform this normalization for bifurcating trees, it is necessary to know the value of $I$ on the rooted caterpillars $K_n$ and the maximally balanced trees $B_n$. It turns out that, while computing $I(K_n)$ is usually easy, to compute $I(B_n)$ one is led in most cases to solve a

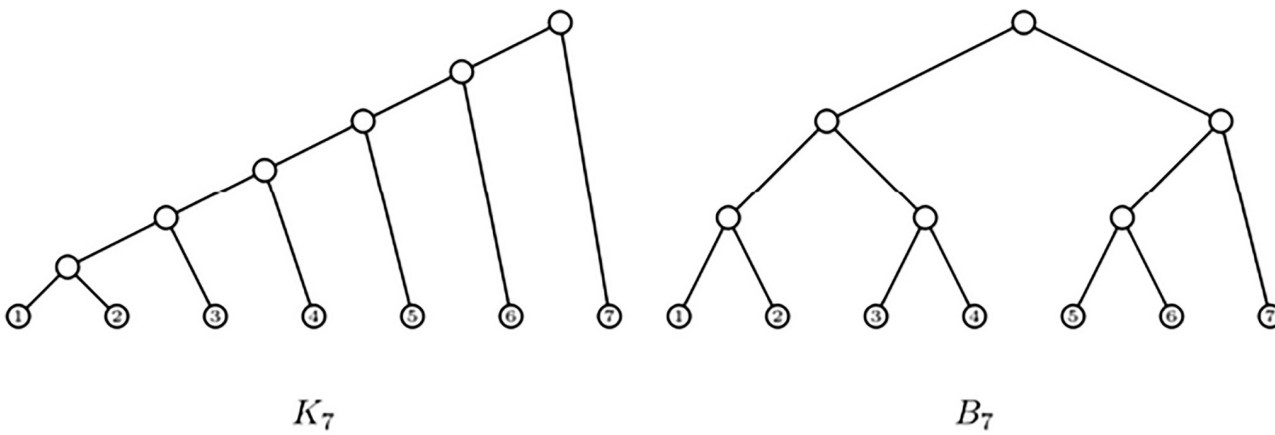

**Fig 1. A rooted caterpillar with 7 leaves (left) and a maximally balanced with 7 leaves (right).**

recurrence of the form

$$I(B_n) = I(B_{\lceil n/2 \rceil}) + I(B_{\lfloor n/2 \rfloor}) + p_I(n)$$

with $p_I(n)$ a function that, in some sense, measures the contribution of the root to the value of $I$. For the four aforementioned indices, this independent term is a polynomial in $\lceil n/2 \rceil$ and $\lfloor n/2 \rfloor$: for the Sackin index, $p_S(n) = n = \lceil n/2 \rceil + \lfloor n/2 \rfloor$; for the Colless index, $p_C(n) = \lceil n/2 \rceil - \lfloor n/2 \rfloor$; for the Total Cophenetic index, $p_\Phi(n) = \binom{n}{2} = \binom{\lceil n/2 \rceil + \lfloor n/2 \rfloor}{2}$; and for the rooted Quartet index, $p_{rQI}(n) = \binom{\lceil n/2 \rceil}{2}\binom{\lfloor n/2 \rfloor}{2}$ [16]. At the moment of writing this paper, the values of $S(B_n)$ and $C(B_n)$ were known (see Examples 2 and 3), but not those of $\Phi(B_n)$ or $rQI(B_n)$.

In the context of the analysis of algorithms, divide-and-conquer recurrences like (1) and more general ones are "solved" by bounding the growing order of $x_n$ using some *Master Theorem*. The original Master Theorem was obtained by Bentley, Haken, and Saxe [19] and extended in [20, §4.5–6], and since then it has been extended to more general divide-and-conquer recurrences: see, for instance [21–28]. Tipically, a Master Theorem deduces information on the asymptotic behaviour of the solution $x_n$ of a divide-and-conquer recurrence from the growing order of the independent term of the recurrence, the number of parts into which the input is divided, which we fix here to 2, and the contribution of each subproblem to the general problem, which we assume here to be equal and constant and represented by the coefficient $a$. More specifically, for the sequences satisfying the Eq (1) considered in this paper, if $d$ is the degree of the bivariate polinomial $p$, the Master Theorems says that (cf. [20, Thm. 4.1])

$$x_n = \begin{cases} \Theta(n^{\log_2(a)}) & \text{if } a > 2^d \\ \Theta(n^{\log_2(a)}\log(n)) & \text{if } a = 2^d \\ \Theta(n^d) & \text{if } a < 2^d \end{cases}$$

Thus, although "it is easy and fast to use" [25, p. 171], the Master Theorem does not provide an explicit solution of the recurrence, only its order of growth.

Now, in the analysis of algorithms, knowing the growing order of the computational cost of an algorithm on an instance of size $n$ is usually enough. But in other applications, like for instance in order to normalize balance indices as we explained above, an explicit expression for the solution is needed. Many specific divide-and-conquer recurrences are explicitly solved when needed, like for instance the cost of the mergesort algorithm in the worst case, solved as

Ex. 34 in [29, Ch. 3], or the homogeneous divide-and-conquer dividing by a half equation (Eq (1) with $p = 0$), solved in [30, Lem. 21]. But no finite expression for the general solution was known so far. To our knowledge, the only attempt to find an explicit solution of Eq (1) is made by Hwang, Janson and Tsai [31] by proving that, when $a = 1$ and under very general conditions on the independent term $p(n)$, the solution $x_n$ has the form

$$x_n = nP(\log_2(n)) + F(n) - Q(n)$$

with $P$ continuous and 1-periodic and $F$, $Q$ of precise growing orders. The authors also give explicit expressions for $P$, $F$, and $Q$ in terms of infinite series expansions, but they are only able to give them in a finite form in a few examples, even for polynomial independent terms.

In this paper we consider the case when $a$ is an arbitrary real number but $p(n)$ is a polynomial in $\lceil n/2 \rceil$ and $\lfloor n/2 \rfloor$. In this case, we are able to give a finitary explicit formula for $x_n$ in terms of the binary decomposition of $n$. Our formula avoids the use of infinite series: actually, the indices involved in all sums in our expression for $x_n$ have a range bounded from above by $\lfloor \log_2(n) \rfloor$ and the degree of the independent term $p(n)$.

The rest of this paper is organized as follows. In the Results section we state our main result and we use it to solve several examples. As a proof of concept, in some of them we obtain formulas that were already known, but we also include several interesting examples for which no explicit formula was known so far. Then we devote a section to outline the proof of our main result; the proofs of the intermediate results are provided in the S1 File. We close the paper with a Conclusions section. We have implemented the formula provided by our main theorem in Python using SymPy, a Python library for symbolic mathematics. This implementation is available at https://github.com/biocom-uib/divide_and_conquer.

## Results

### Main result

For every $n \in \mathbb{N}_{\geqslant 1}$, let its binary decomposition be $n = \sum_{j=1}^{s_n} 2^{q_j(n)}$, with $0 \leqslant q_1(n) < \cdots < q_{s_n}(n)$, and then, for every $i \geqslant 1$, let $M_i(n) = \sum_{j=i}^{s_n} 2^{q_j(n)}$. Notice that $M_1(n) = n$ and $M_{s_n} = 2^{q_{s_n}(n)}$. In order to simplify the notations, we set $q_0(n) = 0$ and $M_0(n) = M_1(n) + 2^{q_0(n)} = n + 1$.

Additionally, for every $d, n \in \mathbb{N}$ and $x \in \mathbb{R}_{\neq 0}$ let

$$T(d, n, x) = \sum_{k=0}^{n-1} k^d x^k$$

(with the convention, which we use throughout this paper, that $0^0 = 1$). In particular,

$$T(0, n, x) = \begin{cases} n & \text{if } x = 1 \\ \dfrac{x^n - 1}{x - 1} & \text{if } x \neq 1 \end{cases} \qquad T(1, n, x) = \begin{cases} \dbinom{n}{2} & \text{if } x = 1 \\ \dfrac{nx^n(x - 1) - x(x^n - 1)}{(x - 1)^2} & \text{if } x \neq 1 \end{cases}$$

For every $m \in \mathbb{N}$, let $B_m$ denote the $m$-th Bernoulli number of the first kind [32]. In particular, $B_0 = 1$, $B_1 = -1/2$, and $B_{2k + 1} = 0$ for every $k \geqslant 1$.

For $d = 0, 1$, for every $a \in \mathbb{R}$, for every $m \in \mathbb{N}$, and for every $n \in \mathbb{N}_{\geqslant 1}$, let

$$\alpha_n^{(d,m)}(a)$$

$$= \frac{1}{2(m+1)} \sum_{i=1}^{s_n} \sum_{j=0}^{m} \binom{m+1}{j} B_j 2^j M_i(n)^{m+1-j} (T(d, q_i(n), a2^{j-m}) - T(d, q_{i-1}(n), a2^{j-m}))$$

$$+ \sum_{i=1}^{s_n-1} q_i(n)^d (a2^{-m})^{q_i(n)} (n - M_i(n)) M_{i+1}(n)^m - T(d, q_{s_n}(n), 2a) \cdot \delta_{m=0}$$

with $\delta_{m=0} = 1$ if $m = 0$ and $\delta_{m=0} = 0$ if $m > 0$.

For every $a \in \mathbb{R}_{>0}$ and for every $r, t \in \mathbb{N}$, let $\ell_{a,t} = \log_2(a) - t$ and

$$\delta_\ell(a, r, t) = \begin{cases} 1 & \text{if } r > 0, \text{ and } \ell_{a,t} \in \{0, \ldots, r-1\} \\ 0 & \text{otherwise} \end{cases}$$

If $a \leqslant 0$, then $\ell_{a,t}$ is undefined and $\delta_\ell(a, r, t) = 0$.

The main result in this paper is the following:

**Theorem 1.** *Let $a \in \mathbb{R}_{\neq 0}$ and let $P(x, y) = \sum_{r,t \geqslant 0} b_{r,t} x^r y^t \in \mathbb{R}[x, y]$ be a bivariate polynomial. Then, the solution of the recurrence equation*

$$x_n = a \cdot x_{\lceil n/2 \rceil} + a \cdot x_{\lfloor n/2 \rfloor} + P(\lceil n/2 \rceil, \lfloor n/2 \rfloor), \quad n \geqslant 2,$$

*with initial condition $x_1$ is*

$$x_n = \sum_{r,t} b_{r,t} x_n^{(r,t)}(a) + ((2a)^{q_{s_n}(n)} + (2a - 1)(na^{q_{s_n}(n)} - (2a)^{q_{s_n}(n)})) x_1$$

*with, for every $n \geqslant 2$,*

$$x_n^{(r,t)}(a) = \sum_{k=1}^{r+t} \left( \sum_{\substack{i=k \\ i \neq t + \ell_{a,t} + 1}}^{r+t} \frac{\binom{r}{i-t-1} \binom{i}{k} B_{i-k}}{i(2^{i-1} - a)} \right) n^k + \frac{1}{a-1} \cdot \delta_{r>0, t=0, a \neq 1}$$

$$+ \left( 1 - \sum_{\substack{l=0 \\ l \neq \ell_{a,t}}}^{r-1} \frac{\binom{r}{l}}{2^{t+l} - a} \right) \left( T(0, q_{s_n}(n), 2a) + na^{q_{s_n}(n)} - (2a)^{q_{s_n}(n)} \right)$$

$$+ \sum_{i=0}^{r+t-1} \left( 2^{-i} \binom{r+t}{i} - 2^{-i+1} \binom{r}{i-t} - \sum_{\substack{l=i-t+1 \\ l \neq \ell_{a,t}}}^{r-1} \frac{\binom{r}{l} \binom{t+l}{i}}{2^{t+l} - a} \right) \alpha_n^{(0,i)}(a)$$

$$+ \frac{\delta_\ell(a, r, t)}{a} \binom{r}{\ell_{a,t}} \left( T(1, q_{s_n}(n), 2a) + (na^{q_{s_n}(n)} - (2a)^{q_{s_n}(n)}) q_{s_n}(n) \right.$$

$$\left. + \sum_{i=0}^{t+\ell_{a,t}-1} \binom{t+\ell_{a,t}}{i} \alpha_n^{(1,i)}(a) \right)$$

*where $\delta_{r>0, t=0, a \neq 1} = 1$ if $r > 0$, $t = 0$, and $a \neq 1$, and $\delta_{r>0, t=0, a \neq 1} = 0$ otherwise.*

In particular:

(a) If $r = 0$,

$$
\begin{aligned}
x_n^{(0,t)}(a) \quad &= \sum_{i=0}^{t-1} 2^{-i}\binom{t}{i}\alpha_n^{(0,i)}(a) + T(0, q_{s_n}(n), 2a) + na^{q_{s_n}(n)} - (2a)^{q_{s_n}(n)} \\
&= \begin{cases}
\displaystyle\sum_{i=0}^{t-1} 2^{-i}\binom{t}{i}\alpha_n^{(0,i)}(1/2) + q_{s_n}(n) + n \cdot 2^{-q_{s_n}(n)} - 1 & \text{if } a = \dfrac{1}{2} \\[2em]
\displaystyle\sum_{i=0}^{t-1} 2^{-i}\binom{t}{i}\alpha_n^{(0,i)}(a) + \dfrac{(2a)^{q_{s_n}(n)} - 1}{2a - 1} + n \cdot a^{q_{s_n}(n)} - (2a)^{q_{s_n}(n)} & \text{if } a \neq \dfrac{1}{2}
\end{cases}
\end{aligned}
$$

(b) If $r \geqslant 1$ and $t = 0$:

(b.1) If $a = 1/2$,

$$
x_n^{(r,0)}(1/2) = 2\sum_{k=1}^{r}\left( \sum_{j=k}^{r} \frac{\binom{r}{j-1}\binom{j}{k}B_{j-k}}{j(2^j - 1)} \right) n^k - 2
$$

$$
+ \left( 1 - 2\sum_{l=0}^{r-1} \frac{\binom{r}{l}}{2^{l+1} - 1} \right)\left( q_{s_n}(n) + n \cdot 2^{-q_{s_n}(n)} - 1 \right)
$$

$$
- \sum_{i=0}^{r-1}\left( 2^{-i}\binom{r}{i} + 2\sum_{l=i+1}^{r-1} \frac{\binom{r}{l}\binom{l}{i}}{2^{l+1} - 1} \right)\alpha_n^{(0,i)}(1/2)
$$

(b.2) If $a = 1$,

$$
x_n^{(r,0)}(1) = \sum_{k=2}^{r}\left( \sum_{j=k}^{r} \frac{\binom{r}{j-1}\binom{j}{k}B_{j-k}}{j(2^{j-1} - 1)} \right) n^k + \left( q_{s_n}(n) + 1 + \sum_{j=1}^{r-1} \frac{\binom{r}{j}(B_j - 1)}{2^j - 1} \right) n
$$

$$
+ 1 + \sum_{j=1}^{r-1} \frac{\binom{r}{j}}{2^j - 1} - 2^{q_{s_n}(n)+1} - \sum_{i=0}^{r-1}\left( 2^{-i}\binom{r}{i} + \sum_{l=i+1}^{r-1} \frac{\binom{r}{l}\binom{l}{i}}{2^l - 1} \right)\alpha_n^{(0,i)}(1)
$$

(b.3) If $a \notin \{1/2, 1, \ldots, 2^{r-1}\}$

$$x_n^{(r,0)}(a) = \sum_{k=1}^{r} \left( \sum_{j=k}^{r} \frac{\binom{r}{j-1}\binom{j}{k}B_{j-k}}{j(2^{j-1}-a)} \right) n^k + \frac{1}{a-1}$$

$$+ \left( 1 - \sum_{l=0}^{r-1} \frac{\binom{r}{l}}{2^l - a} \right) \left( \frac{(2a)^{q_{s_n}(n)} - 1}{2a - 1} + na^{q_{s_n}(n)} - (2a)^{q_{s_n}(n)} \right)$$

$$- \sum_{i=0}^{r-1} \left( 2^{-i}\binom{r}{i} + \sum_{l=i+1}^{r-1} \frac{\binom{r}{l}\binom{l}{i}}{2^l - a} \right) \alpha_n^{(0,i)}(a)$$

(b.4) If $a = 2^\ell$ with $\ell \in \{1, \ldots, r-1\}$,

$$x_n^{(r,0)}(a) = \sum_{k=1}^{r} \left( \sum_{\substack{i=k \\ i \neq \ell+1}}^{r} \frac{\binom{r}{i-1}\binom{i}{k}B_{i-k}}{i(2^{i-1}-a)} \right) n^k + \frac{1}{a-1}$$

$$+ \left( 1 - \sum_{\substack{l=0 \\ l \neq \ell}}^{r-1} \frac{\binom{r}{l}}{2^l - a} \right) \left( \frac{(2a)^{q_{s_n}(n)} - 1}{2a - 1} + na^{q_{s_n}(n)} - (2a)^{q_{s_n}(n)} \right)$$

$$- \sum_{i=0}^{r-1} \left( 2^{-i}\binom{r}{i} + \sum_{\substack{l=i+1 \\ l \neq \ell}}^{r-1} \frac{\binom{r}{l}\binom{l}{i}}{2^l - a} \right) \alpha_n^{(0,i)}(a)$$

$$+ \frac{1}{a}\binom{r}{\ell} \left( \frac{((2a-1)q_{s_n}(n) - 2a)(2a)^{q_{s_n}(n)} + 2a}{(2a-1)^2} + q_{s_n}(n)a^{q_{s_n}(n)}(n - 2^{q_{s_n}(n)}) \right)$$

$$+ \frac{1}{a}\binom{r}{\ell}\sum_{i=0}^{\ell-1}\binom{\ell}{i}\alpha_n^{(1,i)}(a)$$

(c) If $r \geqslant 1$ and $t \geqslant 1$:

(c.1) If $a = 1/2$,

$$x_n^{(r,t)}(1/2) = 2 \sum_{k=1}^{r+t} \left( \sum_{j=k}^{r+t} \frac{\binom{r}{j-t-1}\binom{j}{k} B_{j-k}}{j(2^j - 1)} \right) n^k$$

$$+ \left( 1 - 2 \sum_{l=0}^{r-1} \frac{\binom{r}{l}}{2^{t+l+1} - 1} \right) \left( q_{s_n}(n) + n \cdot 2^{-q_{s_n}(n)} - 1 \right)$$

$$+ \sum_{i=0}^{r+t-1} \left( 2^{-i}\binom{r+t}{i} - 2^{-i+1}\binom{r}{i-t} - 2 \sum_{l=i-t+1}^{r-1} \frac{\binom{r}{l}\binom{t+l}{i}}{2^{t+l+1} - 1} \right) \alpha_n^{(0,i)}(1/2)$$

(c.2) If $a = 1$,

$$x_n^{(r,t)}(1) = \sum_{k=2}^{r+t} \left( \sum_{j=k}^{r+t} \frac{\binom{r}{j-t-1}\binom{j}{k} B_{j-k}}{j(2^{j-1} - 1)} \right) n^k$$

$$+ \left( 1 + \sum_{j=1}^{r+t-1} \frac{\binom{r}{j-t}(B_j - 1)}{2^j - 1} \right) n + \sum_{l=0}^{r-1} \frac{\binom{r}{l}}{2^{t+l} - 1} - 1$$

$$+ \sum_{i=0}^{r+t-1} \left( 2^{-i}\binom{r+t}{i} - 2^{-i+1}\binom{r}{i-t} - \sum_{l=i-t+1}^{r-1} \frac{\binom{r}{l}\binom{t+l}{i}}{2^{t+l} - 1} \right) \alpha_n^{(0,i)}(1)$$

(c.3) If $a \notin \{1/2, 1, 2^t, \ldots, 2^{r+t-1}\}$,

$$x_n^{(r,t)}(a) = \sum_{k=1}^{r+t} \left( \sum_{j=k}^{r+t} \frac{\binom{r}{j-t-1}\binom{j}{k} B_{j-k}}{j(2^{j-1} - a)} \right) n^k$$

$$+ \left( 1 - \sum_{l=0}^{r-1} \frac{\binom{r}{l}}{2^{t+l} - a} \right) \left( \frac{(2a)^{q_{s_n}(n)} - 1}{2a - 1} + na^{q_{s_n}(n)} - (2a)^{q_{s_n}(n)} \right)$$

$$+ \sum_{i=0}^{r+t-1} \left( 2^{-i}\binom{r+t}{i} - 2^{-i+1}\binom{r}{i-t} - \sum_{l=i-t+1}^{r-1} \frac{\binom{r}{l}\binom{t+l}{i}}{2^{t+l} - a} \right) \alpha_n^{(0,i)}(a)$$

(c.4) If $a = 2^{t+\ell}$, for some $\ell \in \{0, \ldots, r-1\}$,

$$
x_n^{(r,t)}(a) = \sum_{k=1}^{r+t} \left( \sum_{\substack{j=k \\ j \neq t+\ell+1}}^{r+t} \frac{\binom{r}{j-t-1}\binom{j}{k}B_{j-k}}{j(2^{j-1}-a)} \right) n^k
$$

$$
+ \frac{1}{a}\binom{r}{\ell}\left( \frac{q_{s_n}(n)(2a)^{q_{s_n}(n)} - (2a)^{q_{s_n}(n)+1} + 2a}{(2a-1)^2} + q_{s_n}(n)a^{q_{s_n}(n)}(n - 2^{q_{s_n}(n)}) \right)
$$

$$
+ \left( 1 - \sum_{\substack{l=0 \\ l \neq \ell}}^{r-1} \frac{\binom{r}{l}}{2^{t+l}-a} \right)\left( \frac{(2a)^{q_{s_n}(n)} - 1}{2a-1} + na^{q_{s_n}(n)} - (2a)^{q_{s_n}(n)} \right)
$$

$$
+ \sum_{i=0}^{r+t-1}\left( 2^{-i}\binom{r+t}{i} - 2^{-i+1}\binom{r}{i-t} - \sum_{\substack{l=i-t+1 \\ l \neq \ell}}^{r-1} \frac{\binom{r}{l}\binom{t+l}{i}}{2^{t+l}-a} \right)\alpha_n^{(0,i)}(a)
$$

$$
+ \frac{1}{a}\binom{r}{\ell}\sum_{i=0}^{t+\ell-1}\binom{t+\ell}{i}\alpha_n^{(1,i)}(a)
$$

## Examples

In this section we gather several applications of our main theorem. The sequences' identifiers, when available, refer to the OEIS.

**Example 1**. When the independent term in the divide-and-conquer recurrence is constant,

$$
x_n = a \cdot x_{\lceil n/2 \rceil} + a \cdot x_{\lfloor n/2 \rfloor} + c
$$

with $c \in \mathbb{R}$, the solution with initial condition $x_1$ is

$$
\begin{aligned}
x_n &= c \cdot x_n^{(0,0)}(a) + ((2a)^{q_{s_n}(n)} + (2a-1)(na^{q_{s_n}(n)} - (2a)^{q_{s_n}(n)}))x_1 \\
&= \begin{cases}
c(q_{s_n}(n) + n \cdot 2^{-q_{s_n}(n)} - 1) + x_1 & \text{if } a = \dfrac{1}{2} \\[2ex]
((2a-1)x_1 + c)\left( na^{q_{s_n}(n)} - \dfrac{2a-2}{2a-1}(2a)^{q_{s_n}(n)} \right) - \dfrac{c}{2a-1} & \text{if } a \neq \dfrac{1}{2}
\end{cases}
\end{aligned}
$$

In particular, if $a = 1$,

$$
x_n = (x_1 + c)n - c.
$$

**Example 2**. The minimum total Sackin index $S_n$ of a rooted bifurcating tree with $n$ leaves (the binary entropy function: sequence A003314) satisfies the recurrence

$$
S_n = S_{\lceil n/2 \rceil} + S_{\lfloor n/2 \rfloor} + n = S_{\lceil n/2 \rceil} + S_{\lfloor n/2 \rfloor} + \lceil n/2 \rceil + \lfloor n/2 \rfloor
$$

with $S_1 = 0$. Then, according to our main theorem,

$$S_n = x_n^{(1,0)}(1) + x_n^{(0,1)}(1)$$

where

$$\begin{aligned} x_n^{(1,0)}(1) &= (q_{s_n}(n) + 1)n + 1 - 2^{q_{s_n}(n)+1} - \alpha_n^{(0,0)}(1) \\ x_n^{(0,1)}(1) &= \alpha_n^{(0,0)}(1) + n - 1 \end{aligned}$$

and therefore

$$\begin{aligned} S_n &= (q_{s_n}(n) + 1)n + 1 - 2^{q_{s_n}(n)+1} - \alpha_n^{(0,0)}(1) + \alpha_n^{(0,0)}(1) + n - 1 \\ &= (q_{s_n}(n) + 2)n - 2^{q_{s_n}(n)+1} \end{aligned}$$

in agreement with previously published formulas: cf. [31, 33].

**Example 3**. The minimum Colless index $c_n$ of a rooted bifurcating tree with $n$ leaves (sequence A296062) satisfies the recurrence

$$c_n = c_{\lceil n/2 \rceil} + c_{\lfloor n/2 \rfloor} + \left\lceil \frac{n}{2} \right\rceil - \left\lfloor \frac{n}{2} \right\rfloor$$

with $c_1 = 0$. Therefore, according to our main theorem

$$\begin{aligned} c_n &= x_n^{(1,0)}(1) - x_n^{(0,1)}(1) \\ &= (q_{s_n}(n) + 1)n + 1 - 2^{q_{s_n}(n)+1} - \alpha_n^{(0,0)}(1) - (\alpha_n^{(0,0)}(1) + n - 1) \\ &= q_{s_n}(n)n - 2^{q_{s_n}(n)+1} + 2 - 2\alpha_n^{(0,0)}(1) \end{aligned}$$

Now,

$$\begin{aligned} \alpha_n^{(0,0)}(1) &= \frac{1}{2} \sum_{i=1}^{s_n} M_i(n)(q_i(n) - q_{i-1}(n)) + \sum_{i=1}^{s_n-1}(n - M_i(n)) - 2^{q_{s_n}(n)} + 1 \\ &= \sum_{i=1}^{s_n} 2^{q_i(n)-1} q_i(n) + (s_n - 1)n - \sum_{j=1}^{s_n}\sum_{i=j}^{s_n} 2^{q_i(n)} + 1 \qquad (2) \\ &= \sum_{i=1}^{s_n} 2^{q_i(n)-1} q_i(n) + (s_n - 1)n - \sum_{i=1}^{s_n} i2^{q_i(n)} + 1 \end{aligned}$$

$$= (s_n - 1)n + \sum_{i=1}^{s_n} 2^{q_i(n)-1}(q_i(n) - 2i) + 1 \qquad (3)$$

and hence, finally,

$$\begin{aligned} c_n &= q_{s_n}(n)n - 2^{q_{s_n}(n)+1} + 2 - 2\Big((s_n - 1)n + \sum_{i=1}^{s_n} 2^{q_i(n)-1}(q_i(n) - 2i) + 1\Big) \\ &= \sum_{i=1}^{s_n-1} 2^{q_i(n)}(q_{s_n}(n) - q_i(n) - 2(s_n - i - 1)) \end{aligned}$$

as it was proved in [34] by showing by induction that this sequence satisfies the recurrence above.

**Example 4**. Consider the equation

$$x_n = x_{\lceil n/2 \rceil} + x_{\lfloor n/2 \rfloor} + (-1)^n$$

This recurrence is related to the Takagi curve [35], which plays an important role in different fields such as analysis, combinatorics, and number theory [36]. Since $(-1)^n = 1 - 2(\lceil n/2 \rceil - \lfloor n/2 \rfloor)$, its solution with initial condition $x_1$ is

$$x_n = x_n^{(0,0)}(1) - 2x_n^{(1,0)}(1) + 2x_n^{(0,1)}(1) + (2^{q_{s_n}(n)} + (2-1)(n \cdot 1^{q_{s_n}(n)} - 2^{q_{s_n}(n)}))x_1$$

$$= n - 1 - 2\sum_{i=1}^{s_n-1} 2^{q_i(n)}(q_{s_n}(n) - q_i(n) - 2(s_n - i - 1)) + nx_1$$

(by Examples 1 and 3)

$$= n - 1 - 2\sum_{i=1}^{s_n-1} 2^{q_i(n)}(q_{s_n}(n) - q_i(n) - 2(s_n - i)) - 4\sum_{i=1}^{s_n-1} 2^{q_i(n)} + nx_1$$

$$= n - 1 - 2\sum_{i=1}^{s_n-1} 2^{q_i(n)}(q_{s_n}(n) - q_i(n) - 2(s_n - i)) - 4(n - 2^{q_{s_n}(n)}) + nx_1$$

$$= 2^{q_{s_n}(n)+2} + (x_1 - 3)n - 1 - 2\sum_{i=1}^{s_n-1} 2^{q_i(n)}(q_{s_n}(n) - q_i(n) - 2(s_n - i))$$

In particular, when $x_1 = 1$,

$$x_n = 2^{q_{s_n}(n)+2} - 2n - 1 - 2\sum_{i=1}^{s_n-1} 2^{q_i(n)}(q_{s_n}(n) - q_i(n) - 2(s_n - i))$$

Consider now the sequence A268289, which we denote by $a_n$. Monroe and Job proved in [37] that

$$a_{n-1} = \sum_{j=0}^{q_{s_n}(n)} [((\lfloor n/2^j \rfloor + 1) \bmod 2)2^j + (-1)^{(\lfloor n/2^j \rfloor + 1) \bmod 2}(n \bmod 2^j)]$$

$$= \sum_{j=0}^{q_{s_n}(n)} 2^j - \sum_{i=1}^{s_n} 2^{q_i(n)} - \sum_{j=0}^{q_{s_n}(n)} \sum_{i \,:\, q_i(n)<j} 2^{q_i(n)} + 2\sum_{j=1}^{s_n} \sum_{i=1}^{j-1} 2^{q_i(n)}$$

$$= 2^{q_{s_n}(n)+1} - 1 - n - \sum_{i=1}^{s_n} 2^{q_i(n)}(q_{s_n}(n) - q_i(n)) + 2\sum_{i=1}^{s_n} 2^{q_i(n)}(s_n - i)$$

$$= 2^{q_{s_n}(n)+1} - 1 - n - \sum_{i=1}^{s_n-1} 2^{q_i(n)}(q_{s_n}(n) - q_i(n) - 2(s_n - i))$$

Using this expression to compute $a_{2n-1}$ (and taking into account that $s_{2n} = s_n$ and $q_i(2n) = q_i(n) + 1$ for $i = 1, \ldots, s_n$) we obtain

$$a_{2n-1} = 2^{q_{s_n}(n)+2} - 1 - 2n - 2\sum_{i=1}^{s_n-1} 2^{q_i(n)}(q_{s_n}(n) - q_i(n) - 2(s_n - i))$$

That is, $x_n = a_{2n-1}$, which is not self-evident from the definitions of both sequences.

**Remark 1**. The solution of

$$x_n = a \cdot x_{\lceil n/2 \rceil} + a \cdot x_{\lfloor n/2 \rfloor} + (\lceil n/2 \rceil - \lfloor n/2 \rfloor)$$

and of

$$x_n = a \cdot x_{\lceil n/2 \rceil} + a \cdot x_{\lfloor n/2 \rfloor} + (\lceil n/2 \rceil - \lfloor n/2 \rfloor)^m$$

for $m \geqslant 1$, is the same. Therefore, for every $m \geqslant 1$,

$$x^{(1,0)}(a) - x^{(0,1)}(a) = \sum_{p=0}^{m} \binom{m}{p} (-1)^{m-p} x^{(p,m-p)}(a)$$

**Remark 2**. The solution of Eq (1) is 0 exactly when $x_1 = 0$ and $p(n) = 0$ for every $n \geqslant 1$. Writing $p(n) = P(\lceil n/2 \rceil, \lfloor n/2 \rfloor)$ with $P(x, y) \in \mathbb{R}[x, y]$, this says that $P(m, m) = P(m + 1, m) = 0$ for every $m \geqslant 1$ and hence that the polynomials $P(x, x)$ and $P(x, x - 1)$ are identically 0, that is, $P(x, y) = (x - y)(x - y - 1)Q(x, y)$ for some $Q(x, y) \in \mathbb{R}[x, y]$. So, $x_n = 0$ for every $n \geqslant 1$ when $x_1 = 0$ and the independent term has the form

$$p(n) = (\lceil n/2 \rceil - \lfloor n/2 \rfloor)(\lceil n/2 \rceil - \lfloor n/2 \rfloor - 1)Q(\lceil n/2 \rceil, \lfloor n/2 \rfloor)$$

with $Q(x, y) \in \mathbb{R}[x, y]$.

Writing $P(x, y) = \sum_{k=0}^{d} \sum_{r=0}^{k} b_{r,k-r} x^r y^{k-r}$ the conditions $P(x, x) = P(x, x - 1) = 0$ are equivalent to the following identities: for every $k = 0, \ldots, d$,

$$\sum_{j=0}^{k} b_{j,k-j} = 0, \quad \sum_{l=1}^{d-k} \sum_{j=0}^{k} \binom{l+j}{j} b_{l+j,k-j} = 0 \tag{4}$$

We deduce that if the coefficients $b_{r,t}$ satisfy Eq (4), and only in this case,

$$\sum_{r,t \geqslant 0} b_{r,t} x^{(r,t)}(a) = 0.$$

This generalizes the identity obtained at the end of the previous remark.

**Example 5**. The sequence $x_n = n^2$ satisfies the equation

$$x_n = 2x_{\lceil n/2 \rceil} + 2x_{\lfloor n/2 \rfloor} + \lfloor n/2 \rfloor - \lceil n/2 \rceil$$

with $x_1 = 1$. Therefore,

$$x_n = x_n^{(0,1)}(2) - x_n^{(1,0)}(2) + 3n2^{q_{s_n}(n)} - 2 \cdot 4^{q_{s_n}(n)}$$

Now,

$$x_n^{(0,1)}(2) = \alpha_n^{(0,0)}(2) + \frac{1}{3}\left(3n \cdot 2^{q_{s_n}(n)} - 2 \cdot 4^{q_{s_n}(n)} - 1\right)$$

$$x_n^{(1,0)}(2) = -n + 1 + \frac{2}{3}\left(3n \cdot 2^{q_{s_n}(n)} - 2 \cdot 4^{q_{s_n}(n)} - 1\right) - \alpha_n^{(0,0)}(2)$$

and

$$\alpha_n^{(0,0)}(2) = \frac{1}{2}\sum_{i=1}^{s_n} M_i(n)\big((2^{q_i(n)} - 1) - (2^{q_i(n)-1} - 1)\big) + \sum_{i=1}^{s_n-1} 2^{q_i(n)}(n - M_i(n)) - \frac{4^{q_i(n)} - 1}{3}$$

$$= \frac{1}{2}\sum_{i=1}^{s_n} M_i(n)(2^{q_i(n)} - 2^{q_{i-1}(n)}) + \sum_{i=1}^{s_n} 2^{q_i(n)}(n - M_i(n)) - \frac{1}{3}(3n2^{q_{s_n}(n)} - 2\cdot 4^{q_{s_n}(n)} - 1)$$

$$= n^2 - \frac{1}{2}\sum_{i=1}^{s_n} M_i(n)2^{q_i(n)} - \frac{1}{2}\sum_{i=1}^{s_n} M_i(n)2^{q_{i-1}(n)} - \frac{1}{3}(3n2^{q_{s_n}(n)} - 2\cdot 4^{q_{s_n}(n)} - 1)$$

$$= n^2 - \frac{1}{2}\sum_{i=1}^{s_n} M_i(n)2^{q_i(n)} - \frac{1}{2}\sum_{i=0}^{s_n} M_{i+1}(n)2^{q_i(n)} - \frac{1}{3}(3n2^{q_{s_n}(n)} - 2\cdot 4^{q_{s_n}(n)} - 1)$$

$$= n^2 - \frac{1}{2}\sum_{i=1}^{s_n} 2^{q_i(n)}(M_i(n) + M_{i+1}(n)) - \frac{1}{2}n - \frac{1}{3}(3n2^{q_{s_n}(n)} - 2\cdot 4^{q_{s_n}(n)} - 1)$$

$$= n^2 - \frac{1}{2}\sum_{i=1}^{s_n} (M_i(n) - M_{i+1}(n))(M_i(n) + M_{i+1}(n)) - \frac{1}{2}n - \frac{1}{3}(3n2^{q_{s_n}(n)} - 2\cdot 4^{q_{s_n}(n)} - 1)$$

$$= n^2 - \frac{1}{2}\sum_{i=1}^{s_n} (M_i(n)^2 - M_{i+1}(n)^2) - \frac{1}{2}n - \frac{1}{3}(3n2^{q_{s_n}(n)} - 2\cdot 4^{q_{s_n}(n)} - 1)$$

$$= \frac{1}{2}n^2 - \frac{1}{2}n - \frac{1}{3}(3n2^{q_{s_n}(n)} - 2\cdot 4^{q_{s_n}(n)} - 1)$$

and therefore

$$x_n = \alpha_n^{(0,0)}(2) + \frac{1}{3}\big(3n\cdot 2^{q_{s_n}(n)} - 2\cdot 4^{q_{s_n}(n)} - 1\big)$$

$$- \left(-n + 1 + \frac{2}{3}\big(3n\cdot 2^{q_{s_n}(n)} - 2\cdot 4^{q_{s_n}(n)} - 1\big) - \alpha_n^{(0,0)}(2)\right) + 3n2^{q_{s_n}(n)} - 2\cdot 4^{q_{s_n}(n)}$$

$$= 2\alpha_n^{(0,0)}(2) + \frac{2}{3}\big(3n\cdot 2^{q_{s_n}(n)} - 2\cdot 4^{q_{s_n}(n)} - 1\big) + n$$

$$= n^2 - n - \frac{2}{3}\big(3n2^{q_{s_n}(n)} - 2\cdot 4^{q_{s_n}(n)} - 1\big) + \frac{2}{3}\big(3n\cdot 2^{q_{s_n}(n)} - 2\cdot 4^{q_{s_n}(n)} - 1\big) + n$$

$$= n^2$$

indeed.

**Example 6**. The Walsh system is a complete orthogonal set of functions that can represent any discrete function in the same sense that trigonometric functions represent any continuous function in Fourier analysis [38]. More specifically, if $r_n(t) = \mathrm{sgn}(\sin(2^n \pi t))$, $n \in \mathbb{N}$, are the Rademacher functions, then the Walsh system $W = (W_k : [0, 1] \to \{-1, 1\})_{k \in \mathbb{N}}$ is defined as follows: for every $t \in [0, 1]$

$$W_0(t) = 1, \quad W_{2^k+i}(t) = r_{n+1}(t)W_i(t), \quad k \geqslant 0, \ 0 \leqslant i \leqslant 2^k - 1.$$

Given an orthogonal system of functions $\varphi = (\varphi_n)_{n \in \mathbb{N}}$ defined on $[a, b]$ and satisfying some technical conditions that are satisfied by the Walsh system [39], the Lebesgue constants $\lambda_n(\varphi)$

of $\varphi$ are defined as

$$\lambda_n(\varphi) = \int_0^1 |\sum_{j=1}^n \varphi_j(t)| \, dt,$$

and they are an important characteristic of $\varphi$.

The sequence $\lambda_n := \lambda_n(W)$ of Lebesgue constants of the Walsh system turns out to satisfy the recurrence

$$\lambda_n = \frac{1}{2}\lambda_{\lceil n/2 \rceil} + \frac{1}{2}\lambda_{\lfloor n/2 \rfloor} + \frac{1}{2}\lceil\frac{n}{2}\rceil - \frac{1}{2}\lfloor\frac{n}{2}\rfloor$$

with $\lambda_1 = 1$ [39, 40]. Therefore,

$$\lambda_n = \frac{1}{2}x_n^{(1,0)}(1/2) - \frac{1}{2}x_n^{(0,1)}(1/2) + 1$$

where

$$x_n^{(1,0)}(1/2) = 2n - q_{s_n}(n) - n \cdot 2^{-q_{s_n}(n)} - 1 - \alpha_n^{(0,0)}(1/2)$$

$$x_n^{(0,1)}(1/2) = \alpha_n^{(0,0)}(1/2) + q_{s_n}(n) + n \cdot 2^{-q_{s_n}(n)} - 1$$

$$\alpha_n^{(0,0)}(1/2)$$

$$= \frac{1}{2}\sum_{i=1}^{s_n} M_i(n)(T(0, q_i(n), 1/2) - T(0, q_{i-1}(n), 1/2)) + \sum_{i=1}^{s_n-1} 2^{-q_i(n)}(n - M_i(n)) - q_{s_n}(n)$$

$$= \sum_{i=1}^{s_n} M_i(n)(2^{-q_{i-1}(n)} - 2^{-q_i(n)}) + \sum_{i=1}^{s_n-1} 2^{-q_i(n)}(n - M_i(n)) - q_{s_n}(n)$$

$$= \sum_{i=1}^{s_n} M_i(n)2^{-q_{i-1}(n)} - \sum_{i=1}^{s_n} M_i(n)2^{-q_i(n)} + \sum_{i=1}^{s_n-1} 2^{-q_i(n)}(n - M_i(n)) - q_{s_n}(n)$$

$$= \sum_{i=0}^{s_n} M_{i+1}(n)2^{-q_i(n)} - \sum_{i=1}^{s_n} M_i(n)2^{-q_i(n)} + \sum_{i=1}^{s_n-1} 2^{-q_i(n)}(n - M_i(n)) - q_{s_n}(n)$$

$$= n + \sum_{i=1}^{s_n} (M_{i+1}(n) - M_i(n))2^{-q_i(n)} + \sum_{i=1}^{s_n-1} 2^{-q_i(n)}(n - M_i(n)) - q_{s_n}(n)$$

$$= n - \sum_{i=1}^{s_n} 2^{q_i(n)} 2^{-q_i(n)} + \sum_{i=1}^{s_n-1} 2^{-q_i(n)}(n - M_i(n)) - q_{s_n}(n)$$

$$= n - s_n + \sum_{i=1}^{s_n-1} 2^{-q_i(n)}(n - M_i(n)) - q_{s_n}(n)$$

Therefore,

$$
\begin{aligned}
\lambda_n &= \frac{1}{2}\left(2n - q_{s_n}(n) - n \cdot 2^{-q_{s_n}(n)} - 1 - \alpha_n^{(0,0)}(1/2)\right) \\
&\quad - \frac{1}{2}\left(\alpha_n^{(0,0)}(1/2) + q_{s_n}(n) + n \cdot 2^{-q_{s_n}(n)} - 1\right) + 1 \\
&= -\alpha_n^{(0,0)}(1/2) + n - q_{s_n}(n) - n \cdot 2^{-q_{s_n}(n)} + 1 \\
&= -n + s_n - \sum_{i=1}^{s_n-1} 2^{-q_i(n)}(n - M_i(n)) + q_{s_n}(n) + n - q_{s_n}(n) - n \cdot 2^{-q_{s_n}(n)} + 1 \\
&= s_n - \sum_{i=1}^{s_n} 2^{-q_i(n)}(n - M_i(n))
\end{aligned}
$$

in agreement with [39, Eqn. (5.5)], where this formula is obtained through an *ad hoc* argument.

**Remark 3**. As R. Stephan points out in [1], the OEIS contains many sequences defined by recurrences of the form

$$
\begin{cases}
a_{2n} = C \cdot a_n + C \cdot a_{n-1} + P(n) \\
a_{2n+1} = 2C \cdot a_n + Q(n)
\end{cases}
$$

for some real number $C$ and functions $P$, $Q$. It is straightforward to check then that the sequence $x_n := a_{n-1}$ satisfies, for $n \geq 2$, the recurrence (cf. [31])

$$
x_n = C \cdot x_{\lceil n/2 \rceil} + C \cdot x_{\lfloor n/2 \rfloor} + Q(\lfloor n/2 \rfloor - 1) + (\lceil n/2 \rceil - \lfloor n/2 \rfloor)(P(\lfloor n/2 \rfloor) - Q(\lfloor n/2 \rfloor - 1)).
$$

If $P$ and $Q$ are polynomials, these sequences are covered by our main theorem.

**Example 7**. In [41], Stanton, Kocay, and Dirksen introduced the sequence $a_n = \sum_{k=1}^{n} \sum_{i=1}^{s_k} (-1)^{q_i(n)}$ (sequence A005536). This sequence is generated by the first Feigenbaum symbolic sequence, of importance in symbolic dynamics [42]. By the second last entry in the last table of [1] and the last remark, the sequence $x_n = a_{n-1}$, for $n \geq 1$, satisfies the recurrence

$$
x_n = -x_{\lceil n/2 \rceil} - x_{\lfloor n/2 \rfloor} + \lfloor n/2 \rfloor
$$

with $x_1 = 0$. Hwang, Janson, and Tsai used this recurrence to obtain the following expression for $x_n$:

$$
f(t) = n\left(\frac{1}{4} + \frac{(-1)^{s_n}}{2}\left(\frac{1}{2} - \frac{2^{1-s_n}}{3}\right) + (-1)^{s_n}\sum_{j \geq 0}(-1)^j 2^{-j-t+\lfloor t \rfloor}\bar{g}(2^{j+t-\lfloor t \rfloor})\right)
$$

where, for every $k \geq 1$, $\bar{g}(k) = (1 - x + \lfloor x \rfloor)/2$ if $\lfloor x \rfloor$ is even and $\bar{g}(k) = (x - \lfloor x \rfloor)/2$ if $\lfloor x \rfloor$ is odd; see [31, Ex 7.1].

Our method produces a simpler expression for $x_n$. Indeed, by our main theorem,

$$
x_n = x_n^{(0,1)}(-1)
$$

where

$$x_n^{(0,1)}(-1) = \alpha_n^{(0,0)}(-1) - \frac{1}{3}((-2)^{q_{s_n}(n)} - 1) + (-1)^{q_{s_n}(n)}n - (-2)^{q_{s_n}(n)}$$

$$= \alpha_n^{(0,0)}(-1) - \frac{4}{3} \cdot (-2)^{q_{s_n}(n)} + \frac{1}{3} + (-1)^{q_{s_n}(n)}n$$

$$\alpha_n^{(0,0)}(-1) = \frac{1}{2}\sum_{i=1}^{s_n}M_i(n)(T(0, q_i(n), -1) - T(0, q_{i-1}(n), -1))$$

$$+ \sum_{i=1}^{s_n-1}(-1)^{q_i(n)}(n - M_i(n)) - T(0, q_{s_n}(n), -2)$$

$$= \frac{1}{4}\sum_{i=1}^{s_n}M_i(n)\left((-1)^{q_{i-1}(n)} - (-1)^{q_i(n)}\right) + \sum_{i=1}^{s_n-1}(-1)^{q_i(n)}(n - M_i(n)) + \frac{1}{3}((-2)^{q_{s_n}(n)} - 1)$$

$$= \frac{1}{4}\sum_{i=1}^{s_n}M_i(n)(-1)^{q_{i-1}(n)} - \frac{1}{4}\sum_{i=1}^{s_n}M_i(n)(-1)^{q_i(n)}) + \sum_{i=1}^{s_n-1}(-1)^{q_i(n)}(n - M_i(n))$$

$$+ \frac{1}{3}((-2)^{q_{s_n}(n)} - 1)$$

$$= \frac{1}{4}\sum_{i=0}^{s_n}M_{i+1}(n)(-1)^{q_i(n)} - \frac{1}{4}\sum_{i=1}^{s_n}M_i(n)(-1)^{q_i(n)} + \sum_{i=1}^{s_n-1}(-1)^{q_i(n)}(n - M_i(n))$$

$$+ \frac{1}{3}((-2)^{q_{s_n}(n)} - 1)$$

$$= \frac{1}{4}n - \frac{1}{4}\sum_{i=1}^{s_n}(-2)^{q_i(n)} + \sum_{i=1}^{s_n-1}(-1)^{q_i(n)}(n - M_i(n)) + \frac{1}{3}((-2)^{q_{s_n}(n)} - 1)$$

and hence

$$x_n = \frac{1}{4}n - \frac{1}{4}\sum_{i=1}^{s_n}(-2)^{q_i(n)} + \sum_{i=1}^{s_n-1}(-1)^{q_i(n)}(n - M_i(n)) + \frac{1}{3}((-2)^{q_{s_n}(n)} - 1)$$

$$- \frac{4}{3} \cdot (-2)^{q_{s_n}(n)} + \frac{1}{3} + (-1)^{q_{s_n}(n)}n$$

$$= \frac{1}{4}n - \frac{1}{4}\sum_{i=1}^{s_n}(-2)^{q_i(n)} + \sum_{i=1}^{s_n}(-1)^{q_i(n)}(n - M_i(n))$$

$$= \frac{1}{4}n - \frac{1}{4}\sum_{i=1}^{s_n}(-2)^{q_i(n)} + \sum_{i=1}^{s_n}\left((-1)^{q_i(n)}\sum_{j=1}^{i-1}2^{q_j(n)}\right)$$

$$= \frac{1}{4}\sum_{i=1}^{s_n}2^{q_i(n)}\left(1 - (-1)^{q_i(n)} + 4\sum_{j=i+1}^{s_n}(-1)^{q_j(n)}\right)$$

**Example 8**. The minimum total cophenetic index $\Phi_n$ (see the Introduction for the definition) of a rooted bifurcating tree with $n$ leaves (sequence A174605) satisfies the recurrence

$$\Phi_n = \Phi_{\lceil n/2 \rceil} + \Phi_{\lfloor n/2 \rfloor} + \binom{\lceil n/2 \rceil}{2} + \binom{\lfloor n/2 \rfloor}{2}$$

$$= \Phi_{\lceil n/2 \rceil} + \Phi_{\lfloor n/2 \rfloor} + \frac{1}{2}\left\lceil\frac{n}{2}\right\rceil^2 - \frac{1}{2}\left\lceil\frac{n}{2}\right\rceil + \frac{1}{2}\left\lfloor\frac{n}{2}\right\rfloor^2 - \frac{1}{2}\left\lfloor\frac{n}{2}\right\rfloor$$

with initial condition $\Phi_1 = 0$ [12]. Therefore

$$\Phi_n = \frac{1}{2}\left(x_n^{(2,0)}(1) + x_n^{(0,2)}(1) - x_n^{(1,0)}(1) - x_n^{(0,1)}(1)\right) = \frac{1}{2}\left(x_n^{(2,0)}(1) + x_n^{(0,2)}(1) - S_n\right)$$

where

$$S_n = x_n^{(1,0)}(1) + x_n^{(0,1)}(1) = (q_{s_n}(n) + 2)n - 2^{q_{s_n}(n)+1}$$

was already computed in Example 2 and

$$
\begin{aligned}
x_n^{(2,0)}(1) &= n^2 + (q_{s_n}(n) - 2)n + 3 - 2^{q_{s_n}(n)+1} - 3\alpha_n^{(0,0)}(1) - \alpha_n^{(0,1)}(1) \\
x_n^{(0,2)}(1) &= \alpha_n^{(0,0)}(1) + \alpha_n^{(0,1)}(1) + n - 1
\end{aligned}
$$

Therefore

$$
\begin{aligned}
\Phi_n &= \frac{1}{2}\Big(n^2 + (q_{s_n}(n) - 2)n + 3 - 2^{q_{s_n}(n)+1} - 3\alpha_n^{(0,0)}(1) - \alpha_n^{(0,1)}(1) \\
&\qquad + \alpha_n^{(0,0)}(1) + \alpha_n^{(0,1)}(1) + n - 1 - ((q_{s_n}(n) + 2)n - 2^{q_{s_n}(n)+1})\Big) \\
&= \frac{1}{2}\left(n^2 - 3n + 2 - 2\alpha_n^{(0,0)}(1)\right) \\
&= \binom{n-1}{2} - \left(s_n n + \sum_{i=1}^{s_n} 2^{q_i(n)-1}(q_i(n) - 2i) - n + 1\right) \\
&\quad \text{(by Eqn. (3))} \\
&= \binom{n}{2} - s_n n - \sum_{i=1}^{s_n} 2^{q_i(n)-1}(q_i(n) - 2i)
\end{aligned}
$$

No closed expression had been published so far for this sequence.

**Example 9**. The maximum rooted quartet index $\rho_n$ (see again the Introduction for the definition) of a rooted bifurcating tree with $n$ leaves (sequence A300445) satisfies the recurrence

$$
\begin{aligned}
\rho_n &= \rho_{\lceil n/2 \rceil} + \rho_{\lfloor n/2 \rfloor} + \binom{\lceil n/2 \rceil}{2}\binom{\lfloor n/2 \rfloor}{2} \\
&= \rho_{\lceil n/2 \rceil} + \rho_{\lfloor n/2 \rfloor} + \frac{1}{4}\left(\left\lceil\frac{n}{2}\right\rceil\left\lfloor\frac{n}{2}\right\rfloor - \left\lceil\frac{n}{2}\right\rceil^2\left\lfloor\frac{n}{2}\right\rfloor - \left\lceil\frac{n}{2}\right\rceil\left\lfloor\frac{n}{2}\right\rfloor^2 + \left\lceil\frac{n}{2}\right\rceil^2\left\lfloor\frac{n}{2}\right\rfloor^2\right)
\end{aligned}
$$

with initial condition $\rho_1 = 0$ [17]. Therefore

$$\rho_n = \frac{1}{4}\left(x_n^{(1,1)}(1) - x_n^{(2,1)}(1) - x_n^{(1,2)}(1) + x_n^{(2,2)}(1)\right)$$

where

$$x_n^{(1,1)}(1) = \frac{1}{2}n^2 + (1 + B_1 - 1)n + (1 - 0 - 1)\alpha_n^{(0,0)}(1) + (1 - 1 - 0)\alpha_n^{(0,1)}(1) = \binom{n}{2}$$

$$x_n^{(2,1)}(1) = \sum_{k=2}^{3}\left(\sum_{j=k}^{3}\frac{\binom{2}{j-2}\binom{j}{k}B_{j-k}}{j(2^{j-1}-1)}\right)n^k + \left(1 + \sum_{l=1}^{2}\frac{\binom{2}{l-1}(B_l - 1)}{2^l - 1}\right)n + \sum_{l=0}^{1}\frac{\binom{2}{l}}{2^{1+l}-1} - 1$$

$$+ \sum_{i=0}^{2}\left(2^{-i}\binom{3}{i} - 2^{-i+1}\binom{2}{i-1} - \sum_{l=i}^{1}\frac{\binom{2}{l}\binom{1+l}{i}}{2^{1+l}-1}\right)\alpha_n^{(0,i)}(1)$$

$$= \frac{2}{9}n^3 + \frac{1}{6}n^2 - \frac{19}{18}n + \frac{2}{3} - \frac{2}{3}\alpha_n^{(0,0)}(1) - \frac{5}{6}\alpha_n^{(0,1)}(1) - \frac{1}{4}\alpha_n^{(0,2)}(1)$$

$$x_n^{(1,2)} = \sum_{k=2}^{3}\left(\sum_{j=k}^{3}\frac{\binom{1}{j-3}\binom{j}{k}B_{j-k}}{j(2^{j-1}-1)}\right)n^k + \left(1 + \sum_{l=1}^{2}\frac{\binom{1}{l-2}(B_l - 1)}{2^l - 1}\right)n + \sum_{l=0}^{0}\frac{\binom{1}{l}}{2^{2+l}-1} - 1$$

$$+ \sum_{i=0}^{2}\left(2^{-i}\binom{3}{i} - 2^{-i+1}\binom{1}{i-2} - \sum_{l=i-1}^{0}\frac{\binom{1}{l}\binom{2+l}{i}}{2^{2+l}-1}\right)\alpha_n^{(0,i)}(1)$$

$$= \frac{1}{9}n^3 - \frac{1}{6}n^2 + \frac{13}{18}n - \frac{2}{3} + \frac{2}{3}\alpha_n^{(0,0)}(1) + \frac{5}{6}\alpha_n^{(0,1)}(1) + \frac{1}{4}\alpha_n^{(0,2)}(1)$$

$$x_n^{(2,2)} = \sum_{k=2}^{4}\left(\sum_{j=k}^{4}\frac{\binom{2}{j-3}\binom{j}{k}B_{j-k}}{j(2^{j-1}-1)}\right)n^k + \left(1 + \sum_{l=1}^{3}\frac{\binom{2}{l-2}(B_l - 1)}{2^l - 1}\right)n + \sum_{l=0}^{1}\frac{\binom{2}{l}}{2^{2+l}-1} - 1$$

$$+ \sum_{i=0}^{3}\left(2^{-i}\binom{4}{i} - 2^{-i+1}\binom{2}{i-2} - \sum_{l=i-1}^{1}\frac{\binom{2}{l}\binom{2+l}{i}}{2^{2+l}-1}\right)\alpha_n^{(0,i)}(1)$$

$$= \frac{1}{14}n^4 - \frac{2}{63}n^3 - \frac{2}{21}n^2 + \frac{55}{126}n - \frac{8}{21} + \frac{8}{21}\alpha_n^{(0,0)}(1) + \frac{10}{21}\alpha_n^{(0,1)}(1) + \frac{1}{7}\alpha_n^{(0,2)}(1)$$

Therefore,

$$\begin{aligned}
\rho_n &= \frac{1}{4}\left[\binom{n}{2} - \left(\frac{2}{9}n^3 + \frac{1}{6}n^2 - \frac{19}{18}n + \frac{2}{3} - \frac{2}{3}\alpha_n^{(0,0)}(1) - \frac{5}{6}\alpha_n^{(0,1)}(1) - \frac{1}{4}\alpha_n^{(0,2)}(1)\right)\right.\\
&\quad - \left(\frac{1}{9}n^3 - \frac{1}{6}n^2 + \frac{13}{18}n - \frac{2}{3} + \frac{2}{3}\alpha_n^{(0,0)}(1) + \frac{5}{6}\alpha_n^{(0,1)}(1) + \frac{1}{4}\alpha_n^{(0,2)}(1)\right)\\
&\quad \left. + \left(\frac{1}{14}n^4 - \frac{2}{63}n^3 - \frac{2}{21}n^2 + \frac{55}{126}n - \frac{8}{21} + \frac{8}{21}\alpha_n^{(0,0)}(1) + \frac{10}{21}\alpha_n^{(0,1)}(1) + \frac{1}{7}\alpha_n^{(0,2)}(1)\right)\right]\\
&= \frac{1}{504}\left[(n-3)(n-2)(n-1)(9n+8) + 48\alpha_n^{(0,0)}(1) + 60\alpha_n^{(0,1)}(1) + 18\alpha_n^{(0,2)}(1)\right]
\end{aligned}$$

(5)

Now,

$$\alpha_n^{(0,0)}(1) = \frac{1}{2}\sum_{i=1}^{s_n} M_i(n)(q_i(n) - q_{i-1}(n)) + \sum_{i=1}^{s_n-1}(n - M_i(n)) - 2^{q_{s_n}(n)} + 1 \quad \text{(by Eqn. (2))}$$

$$= \frac{1}{2}\sum_{i=1}^{s_n} M_i(n)(q_i(n) - q_{i-1}(n)) + \sum_{i=1}^{s_n}(n - M_i(n)) - n + 1$$

$$\alpha_n^{(0,1)}(1) = \frac{1}{4}\sum_{i=1}^{s_n}\left(M_i(n)^2(T(0, q_i(n), 1/2) - T(0, q_{i-1}(n), 1/2)) + 4B_1 M_i(n)(q_i(n) - q_{i-1}(n))\right)$$

$$+ \sum_{i=1}^{s_n-1} 2^{-q_i(n)}(n - M_i(n))M_{i+1}(n)$$

$$= \frac{1}{2}\sum_{i=1}^{s_n} M_i(n)^2\left(2^{-q_{i-1}(n)} - 2^{-q_i(n)}\right) - \frac{1}{2}\sum_{i=1}^{s_n} M_i(n)(q_i(n) - q_{i-1}(n)) + \sum_{i=1}^{s_n} 2^{-q_i(n)}M_{i+1}(n)$$

$$\alpha_n^{(0,2)}(1) = \frac{1}{6}\sum_{i=1}^{s_n}\left(M_i(n)^3(T(0, q_i(n), 1/4) - T(0, q_{i-1}(n), 1/4))\right.$$

$$+ 6B_1 M_i(n)^2(T(0, q_i(n), 1/2) - T(0, q_{i-1}(n), 1/2)) + 12B_2 M_i(n)(q_i(n) - q_{i-1}(n)))$$

$$+ \sum_{i=1}^{s_n-1} 4^{-q_i(n)}(n - M_i(n))M_{i+1}(n)^2$$

$$= \frac{2}{9}\sum_{i=1}^{s_n} M_i(n)^3(4^{-q_{i-1}(n)} - 4^{-q_i(n)}) - \sum_{i=1}^{s_n} M_i(n)^2(2^{-q_{i-1}(n)} - 2^{-q_i(n)})$$

$$+ \frac{1}{3}\sum_{i=1}^{s_n} M_i(n)(q_i - q_{i-1}) + \sum_{i=1}^{s_n} 4^{-q_i(n)}M_{i+1}(n)^2(n - M_i(n))$$

and then, substituting these expressions in Eq (5), we finally obtain

$$\rho_n = \frac{1}{252}\left[3(9n + 8)\binom{n-1}{3} - 24n + 24\right.$$

$$+ 2\sum_{i=1}^{s_n} M_i(n)^3(4^{-q_{i-1}(n)} - 4^{-q_i(n)}) + 6\sum_{i=1}^{s_n} M_i(n)^2(2^{-q_{i-1}(n)} - 2^{-q_i(n)})$$

$$+ \sum_{i=1}^{s_n}(n - M_i(n))(24 + 30 \cdot 2^{-q_i(n)}M_{i+1}(n) + 9 \cdot 4^{-q_i(n)}M_{i+1}(n)^2)\Big]$$

This expression can be simplified as follows. Notice that

$$2\sum_{i=1}^{s_n}M_i(n)^3(4^{-q_{i-1}(n)}-4^{-q_i(n)})+6\sum_{i=1}^{s_n}M_i(n)^2(2^{-q_{i-1}(n)}-2^{-q_i(n)})$$

$$=2\left(\sum_{i=0}^{s_n}M_{i+1}(n)^34^{-q_i(n)}-\sum_{i=1}^{s_n}M_i(n)^34^{-q_i(n)}\right)$$

$$+6\left(\sum_{i=0}^{s_n-1}M_{i+1}(n)^22^{-q_i(n)}-\sum_{i=1}^{s_n}M_i(n)^22^{-q_i(n)}\right)$$

$$=2n^3+2\sum_{i=1}^{s_n}4^{-q_i(n)}(M_{i+1}(n)^3-M_i(n)^3)$$

$$+6n^2+6\sum_{i=1}^{s_n}2^{-q_i(n)}(M_{i+1}(n)^2-M_i(n)^2)$$

$$=2n^3+6n^2+2\sum_{i=1}^{s_n}4^{-q_i(n)}(-3M_i(n)^22^{q_i(n)}+3M_i(n)4^{q_i(n)}-8^{q_i(n)})$$

$$+6\sum_{i=1}^{s_n}2^{-q_i(n)}(-2M_i(n)2^{q_i(n)}+4^{q_i(n)})$$

$$=2n^3+6n^2-2\sum_{i=1}^{s_n}(3M_i(n)^22^{-q_i(n)}+3M_i(n)-2\cdot2^{q_i(n)})$$

$$=2n^3+6n^2+4n-6\sum_{i=1}^{s_n}M_i(n)(M_i(n)2^{-q_i(n)}+1)$$

and

$$\sum_{i=1}^{s_n}(n-M_i(n))(24+30\cdot2^{-q_i(n)}M_{i+1}(n)+9\cdot4^{-q_i(n)}M_{i+1}(n)^2)$$

$$=\sum_{i=1}^{s_n}(n-M_i(n))(24+30\cdot2^{-q_i(n)}(M_i(n)-2^{q_i(n)})+9\cdot4^{-q_i(n)}(M_i(n)-2^{q_i(n)})^2)$$

$$=\sum_{i=1}^{s_n}(n-M_i(n))(3+12\cdot2^{-q_i(n)}M_i(n)+9\cdot4^{-q_i(n)}M_i(n)^2)$$

$$=3n\sum_{i=1}^{s_n}(1+4\cdot2^{-q_i(n)}M_i(n)+3\cdot4^{-q_i(n)}M_i(n)^2)$$

$$-3\sum_{i=1}^{s_n}M_i(n)(1+4\cdot2^{-q_i(n)}M_i(n)+3\cdot4^{-q_i(n)}M_i(n)^2)$$

and hence, finally

$$\rho_n=\frac{1}{252}\left[3(9n+8)\binom{n-1}{3}-24n+24+2n^3+6n^2+4n\right.$$

$$-6\sum_{i=1}^{s_n}M_i(n)(M_i(n)2^{-q_i(n)}+1)+3n\sum_{i=1}^{s_n}(1+4\cdot2^{-q_i(n)}M_i(n)+3\cdot2^{-2q_i(n)}M_i(n)^2)$$

$$\left.-3\sum_{i=1}^{s_n}M_i(n)(1+4\cdot2^{-q_i(n)}M_i(n)+3\cdot2^{-2q_i(n)}M_i(n)^2)\right]$$

$$=\frac{1}{504}[9n^4-42n^3+63n^2-6n+6n\sum_{i=1}^{s_n}(1+M_i(n)2^{-q_i(n)})(1+3M_i(n)2^{-q_i(n)})$$

$$-18\sum_{i=1}^{s_n}M_i(n)(1+M_i(n)2^{-q_i(n)})^2]$$

where notice that $M_i(n)2^{-q_i(n)} = n/2^{q_i(n)}\rfloor$. Again, no closed expression was known so far for this sequence, either.

**Example 10**. Let $a_{1,n}$, $a_{2,n}$, and $a_{3,n}$ denote the sequences A006581, A006582, and A006583, respectively. They are defined as

$$a_{1,n} = \sum_{k=1}^{n-1}(k \text{ AND } (n-k)), \quad a_{2,n} = \sum_{k=1}^{n-1}(k \text{ XOR } (n-k)), \quad a_{3,n} = \sum_{k=1}^{n-1}(k \text{ OR } (n-k))$$

where the bitwise operations AND, XOR, and OR are performed on the binary representations of the numbers. For instance,

$$
\begin{aligned}
a_{1,5} &= \sum_{k=1}^{4}(k \text{ AND } (n-k)) = (1 \text{ AND } 4) + (2 \text{ AND } 3) + (3 \text{ AND } 2) + (4 \text{ AND } 1) \\
&= (001 \text{ AND } 100) + (010 \text{ AND } 011) + (011 \text{ AND } 010) + (100 \text{ AND } 001) \\
&= 000 + 010 + 010 + 000 = 0 + 2 + 2 + 0 = 4 \\
a_{2,5} &= \sum_{k=1}^{4}(k \text{ XOR } (n-k)) = (1 \text{ XOR } 4) + (2 \text{ XOR } 3) + (3 \text{ XOR } 2) + (4 \text{ XOR } 1) \\
&= (001 \text{ XOR } 100) + (010 \text{ XOR } 011) + (011 \text{ XOR } 010) + (100 \text{ XOR } 001) \\
&= 101 + 001 + 001 + 101 = 5 + 1 + 1 + 5 = 12 \\
a_{3,5} &= \sum_{k=1}^{4}(k \text{ OR } (n-k)) = (1 \text{ OR } 4) + (2 \text{ OR } 3) + (3 \text{ OR } 2) + (4 \text{ OR } 1) \\
&= (001 \text{ OR } 100) + (010 \text{ OR } 011) + (011 \text{ OR } 010) + (100 \text{ OR } 001) \\
&= 101 + 011 + 011 + 101 = 5 + 3 + 3 + 5 = 16
\end{aligned}
$$

To our knowledge, no explicit formulas for these sequences had been published so far. On the other hand, it is clear from the definition that $a_{1,n} + a_{2,n} = a_{3,n}$, and therefore it is enough to obtain explicit formulas for two of them. We shall focus here on $a_{1,n}$ and $a_{3,n}$.

By Remark 3, the sequences $\sigma_{i,n} = a_{i,n-1}$, for $i = 1, 3$, are the solutions with $\sigma_{1,1} = \sigma_{3,1} = 0$ of the recurrences

$$\sigma_{i,n} = 2\sigma_{i,\lceil n/2 \rceil} + 2\sigma_{i,\lfloor n/2 \rfloor} + g_i(n)$$

where

$$
\begin{aligned}
g_1(n) &= \left\lfloor \frac{n}{2} \right\rfloor \left( \left\lceil \frac{n}{2} \right\rceil - \left\lfloor \frac{n}{2} \right\rfloor \right) = \left\lfloor \frac{n}{2} \right\rfloor \left\lceil \frac{n}{2} \right\rceil - \left\lfloor \frac{n}{2} \right\rfloor^2 \\
g_3(n) &= 6\left( \left\lfloor \frac{n}{2} \right\rfloor - 1 \right) + \left( \left\lceil \frac{n}{2} \right\rceil - \left\lfloor \frac{n}{2} \right\rfloor \right)\left( 5\left\lfloor \frac{n}{2} \right\rfloor - 4 - 6\left\lfloor \frac{n}{2} \right\rfloor + 6 \right) \\
&= 4\left\lfloor \frac{n}{2} \right\rfloor + 2\left\lceil \frac{n}{2} \right\rceil - 6 - g_1(n)
\end{aligned}
$$

Therefore,

$$
\begin{aligned}
\sigma_{1,n} &= x_n^{(1,1)}(2) - x_n^{(0,2)}(2) \\
\sigma_{3,n} &= 4x_n^{(0,1)}(2) + 2x_n^{(1,0)}(2) - 6x_n^{(0,0)}(2) - \sigma_{1,n}
\end{aligned}
$$

where

$$x_n^{(0,0)}(2) = \frac{1}{3}\left(3n \cdot 2^{q_{s_n}(n)} - 2 \cdot 4^{q_{s_n}(n)} - 1\right) \quad \text{(cf. Example 1)}$$

$$x_n^{(0,1)}(2) = \alpha_n^{(0,0)}(2) + x_n^{(0,0)}(2)$$

$$x_n^{(0,2)}(2) = \alpha_n^{(0,0)}(2) + \alpha_n^{(0,1)}(2) + x_n^{(0,0)}(2)$$

$$x_n^{(1,0)}(2) = -n + \frac{2}{3}\left(3n \cdot 2^{q_{s_n}(n)} - 2 \cdot 4^{q_{s_n}(n)} - 1\right) + 1 - \alpha_n^{(0,0)}(2)$$

$$= 2x_n^{(0,0)}(2) - n + 1 - \alpha_n^{(0,0)}(2)$$

$$
\begin{aligned}
x_n^{(1,1)}(2) = & \sum_{k=1}^{2}\left(\sum_{\substack{j=k \\ j \neq 2}}^{2} \frac{\binom{1}{j-2}\binom{j}{k}B_{j-k}}{j(2^{j-1}-2)}\right)n^k \\
& + \frac{1}{2}\left(\frac{(q_{s_n}(n)-1)4^{q_{s_n}(n)+1} - q_{s_n}(n)4^{q_{s_n}(n)} + 4}{9} + nq_{s_n}(n)2^{q_{s_n}(n)} - q_{s_n}(n)4^{q_{s_n}(n)}\right) \\
& + \left(1 - \sum_{\substack{l=0 \\ l \neq 0}}^{0} \frac{\binom{1}{l}}{2^{1+l}-2}\right)\frac{1}{3}\left(3n \cdot 2^{q_{s_n}(n)} - 2 \cdot 4^{q_{s_n}(n)} - 1\right) \\
& + \sum_{i=0}^{1}\left(2^{-i}\binom{2}{i} - 2^{-i+1}\binom{1}{i-1} - \sum_{\substack{l=i \\ l \neq 0}}^{0}\frac{\binom{1}{l}\binom{1+l}{i}}{2^{1+l}-2}\right)\alpha_n^{(0,i)}(2) + \frac{1}{2}\sum_{i=0}^{0}\binom{1}{i}\alpha_n^{(1,i)}(a) \\
= & \frac{1}{18}\left(9nq_{s_n}(n)2^{q_{s_n}(n)} - 6q_{s_n}(n)4^{q_{s_n}(n)} - 4 \cdot 4^{q_{s_n}(n)} + 4\right) \\
& + \frac{1}{3}\left(3n \cdot 2^{q_{s_n}(n)} - 2 \cdot 4^{q_{s_n}(n)} - 1\right) + \alpha_n^{(0,0)}(2) + \frac{1}{2}\alpha_n^{(1,0)}(2)
\end{aligned}
$$

and

$$
\begin{aligned}
\alpha_n^{(0,0)}(2) = & \frac{1}{2}\sum_{i=1}^{s_n}M_i(n)(T(0,q_i(n),2) - T(0,q_{i-1}(n),2)) + \sum_{i=1}^{s_n-1}2^{q_i(n)}(n - M_i(n)) - T(0,q_{s_n}(n),4) \\
= & \frac{1}{2}\sum_{i=1}^{s_n}M_i(n)(2^{q_i(n)} - 2^{q_{i-1}(n)}) + \sum_{i=1}^{s_n-1}2^{q_i(n)}(n - M_i(n)) - \frac{1}{3}(4^{q_{s_n}(n)} - 1)
\end{aligned}
$$

$$
\begin{aligned}
\alpha_n^{(0,1)}(2) = & \frac{1}{4}\sum_{i=1}^{s_n}M_i(n)^2(q_i(n) - q_{i-1}(n)) - \frac{1}{2}\sum_{i=1}^{s_n}M_i(n)(2^{q_i(n)} - 2^{q_{i-1}(n)}) \\
& + \sum_{i=1}^{s_n-1}(n - M_i(n))M_{i+1}(n)
\end{aligned}
$$

$$
\begin{aligned}
\alpha_n^{(1,0)}(2) = & \frac{1}{2}\sum_{i=1}^{s_n}M_i(n)(T(1,q_i(n),2) - T(1,q_{i-1}(n),2)) + \sum_{i=1}^{s_n-1}q_i(n)2^{q_i(n)}(n - M_i(n)) \\
& - T(1,q_{s_n}(n),4) \\
= & \frac{1}{2}\sum_{i=1}^{s_n}M_i(n)((q_i(n)-2)2^{q_i(n)} - (q_{i-1}(n)-2)2^{q_{i-1}(n)}) \\
& + \sum_{i=1}^{s_n-1}q_i(n)2^{q_i(n)}(n - M_i(n)) - \frac{1}{9}\left((3q_{s_n}(n)-4)4^{q_{s_n}(n)} + 4\right)
\end{aligned}
$$

Therefore, finally,

$$
\begin{aligned}
\sigma_{1,n} &= x_n^{(1,1)}(2) - x_n^{(0,2)}(2) \\
&= \frac{1}{18}\left(9nq_{s_n}(n)2^{q_{s_n}(n)} - 6q_{s_n}(n)4^{q_{s_n}(n)} - 4 \cdot 4^{q_{s_n}(n)} + 4\right) \\
&\quad + \frac{1}{3}\left(3n \cdot 2^{q_{s_n}(n)} - 2 \cdot 4^{q_{s_n}(n)} - 1\right) + \alpha_n^{(0,0)}(2) + \frac{1}{2}\alpha_n^{(1,0)}(2) \\
&\quad - \alpha_n^{(0,0)}(2) - \alpha_n^{(0,1)}(2) - \frac{1}{3}\left(3n \cdot 2^{q_{s_n}(n)} - 2 \cdot 4^{q_{s_n}(n)} - 1\right) \\
&= \frac{1}{18}\left(9nq_{s_n}(n)2^{q_{s_n}(n)} - 6q_{s_n}(n)4^{q_{s_n}(n)} - 4 \cdot 4^{q_{s_n}(n)} + 4\right) + \frac{1}{2}\alpha_n^{(1,0)}(2) - \alpha_n^{(0,1)}(2) \\
&= \frac{1}{18}\left(9nq_{s_n}(n)2^{q_{s_n}(n)} - 6q_{s_n}(n)4^{q_{s_n}(n)} - 4 \cdot 4^{q_{s_n}(n)} + 4\right) \\
&\quad + \frac{1}{4}\sum_{i=1}^{s_n}M_i(n)\left((q_i(n)-2)2^{q_i(n)} - (q_{i-1}(n)-2)2^{q_{i-1}(n)}\right) \\
&\quad + \frac{1}{2}\sum_{i=1}^{s_n-1}q_i(n)2^{q_i(n)}(n - M_i(n)) - \frac{1}{18}\left(3q_{s_n}(n)4^{q_{s_n}(n)} - 4 \cdot 4^{q_{s_n}(n)} + 4\right) \\
&\quad - \frac{1}{4}\sum_{i=1}^{s_n}M_i(n)^2(q_i(n) - q_{i-1}(n)) + \frac{1}{2}\sum_{i=1}^{s_n}M_i(n)(2^{q_i(n)} - 2^{q_{i-1}(n)}) \\
&\quad - \sum_{i=1}^{s_n-1}(n - M_i(n))M_{i+1}(n) \\
&= \frac{1}{2}\left(nq_{s_n}(n)2^{q_{s_n}(n)} - q_{s_n}(n)4^{q_{s_n}(n)}\right) + \frac{1}{4}\sum_{i=1}^{s_n}M_i(n)\left(q_i(n)2^{q_i(n)} - q_{i-1}(n)2^{q_{i-1}(n)}\right) \\
&\quad - \frac{1}{4}\sum_{i=1}^{s_n}M_i(n)^2(q_i(n) - q_{i-1}(n)) + \frac{1}{2}\sum_{i=1}^{s_n-1}q_i(n)2^{q_i(n)}(n - M_i(n)) \\
&\quad - \sum_{i=1}^{s_n-1}(n - M_i(n))M_{i+1}(n) \\
&= \frac{1}{4}\sum_{i=1}^{s_n}M_i(n)\left(q_i(n)2^{q_i(n)} - q_{i-1}(n)2^{q_{i-1}(n)}\right) - \frac{1}{4}\sum_{i=1}^{s_n}M_i(n)^2(q_i(n) - q_{i-1}(n)) \\
&\quad + \frac{1}{2}\sum_{i=1}^{s_n}q_i(n)2^{q_i(n)}(n - M_i(n)) - \sum_{i=1}^{s_n}(n - M_i(n))M_{i+1}(n) \\
&= \frac{1}{4}\sum_{i=1}^{s_n}M_i(n)q_i(n)(2^{q_i(n)} - M_i(n)) + \frac{1}{4}\sum_{i=1}^{s_n}M_i(n)q_{i-1}(n)(M_i(n) + 2^{q_{i-1}(n)}) \\
&\quad - \frac{1}{2}\sum_{i=1}^{s_n}M_i(n)q_{i-1}(n)2^{q_{i-1}(n)} + \frac{1}{2}\sum_{i=1}^{s_n}q_i(n)2^{q_i(n)}(n - M_{i+1}(n) - 2^{q_i(n)}) \\
&\quad - \sum_{i=1}^{s_n}(n - M_i(n))M_{i+1}(n) \\
&= -\frac{1}{4}\sum_{i=1}^{s_n}M_i(n)q_i(n)M_{i+1}(n) + \frac{1}{4}\sum_{i=1}^{s_n}M_i(n)q_{i-1}(n)M_{i-1}(n) \\
&\quad - \frac{1}{2}\sum_{i=0}^{s_n}M_{i+1}(n)q_i(n)2^{q_i(n)} - \frac{1}{2}\sum_{i=1}^{s_n}q_i(n)2^{q_i(n)}M_{i+1}(n) \\
&\quad + \frac{1}{2}\sum_{i=1}^{s_n}q_i(n)2^{q_i(n)}(n - 2^{q_i(n)}) - \sum_{i=1}^{s_n}(n - M_i(n))M_{i+1}(n) \\
&= -\frac{1}{4}\sum_{i=1}^{s_n}M_i(n)q_i(n)M_{i+1}(n) + \frac{1}{4}\sum_{i=0}^{s_n-1}M_{i+1}(n)q_i(n)M_i(n) \\
&\quad - \sum_{i=1}^{s_n}q_i(n)2^{q_i(n)}M_{i+1}(n) + \frac{1}{2}\sum_{i=1}^{s_n}q_i(n)2^{q_i(n)}(n - 2^{q_i(n)}) - \sum_{i=1}^{s_n}(n - M_i(n))M_{i+1}(n) \\
&= \frac{1}{2}\sum_{i=1}^{s_n}q_i(n)2^{q_i(n)}(n - 2^{q_i(n)} - 2M_{i+1}(n)) - \sum_{i=1}^{s_n}(n - M_i(n))M_{i+1}(n)
\end{aligned}
$$

and

$$
\begin{aligned}
\sigma_{3,n} &= 4x_n^{(0,1)}(2) + 2x_n^{(1,0)}(2) - 6x_n^{(0,0)}(2) - \sigma_{1,n} \\
&= 4(\alpha_n^{(0,0)}(2) + x_n^{(0,0)}(2)) + 2(2x_n^{(0,0)}(2) - n + 1 - \alpha_n^{(0,0)}(2)) - 6x_n^{(0,0)}(2) - \sigma_{1,n} \\
&= 2\alpha_n^{(0,0)}(2) + 2x_n^{(0,0)}(2)) - 2n + 2 - \sigma_{1,n} \\
&= \sum_{i=1}^{s_n} M_i(n)(2^{q_i(n)} - 2^{q_{i-1}(n)}) + 2\sum_{i=1}^{s_n-1} 2^{q_i(n)}(n - M_i(n)) - \frac{2}{3}(4^{q_{s_n}(n)} - 1) \\
&\quad + \frac{2}{3}(3n \cdot 2^{q_{s_n}(n)} - 2 \cdot 4^{q_{s_n}(n)} - 1) - 2n + 2 - \sigma_{1,n} \\
&= \sum_{i=1}^{s_n} M_i(n)(2^{q_i(n)} - 2^{q_{i-1}(n)}) + 2\sum_{i=1}^{s_n} 2^{q_i(n)}(n - M_i(n)) - 2n + 2 - \sigma_{1,n} \\
&= -\sum_{i=1}^{s_n} M_i(n)2^{q_{i-1}(n)} + n^2 + \sum_{i=1}^{s_n} 2^{q_i(n)}(n - M_i(n)) - 2n + 2 - \sigma_{1,n} \\
&= 2\binom{n-1}{2} - \sigma_{1,n}
\end{aligned}
$$

Therefore,

$$
\begin{aligned}
a_{1,n} &= \frac{1}{2}\sum_{i=1}^{s_{n+1}} q_i(n+1)2^{q_i(n+1)}(n+1 - 2^{q_i(n+1)} - 2M_{i+1}(n+1)) \\
&\qquad\qquad - \sum_{i=1}^{s_{n+1}}(n+1 - M_i(n+1))M_{i+1}(n+1) \\
a_{3,n} &= 2\binom{n}{2} - a_{1,n} \\
a_{2,n} &= a_{3,n} - a_{1,n} = 2\binom{n}{2} - 2a_{1,n}
\end{aligned}
$$

**Example 11**. Notice that $x_{2^m}^{(r,t)} = 2ax_{2^{m-1}}^{(r,t)} + 2^{(m-1)(r+t)}$ and therefore, since $x_1^{(r,t)} = 0$,

$$
\begin{aligned}
x_{2^m}^{(r,t)}(a) &= \sum_{k=0}^{m-1}(2a)^k 2^{(r+t)(m-k-1)} = 2^{(r+t)(m-1)}\sum_{k=0}^{m-1}(2^{-r-t+1}a)^k \\
&= 2^{(r+t)(m-1)}T(0, m, 2^{-r-t+1}a) = \begin{cases} 2^{(r+t)(m-1)}m & \text{if } a = 2^{r+t-1}, \text{ i.e., if } \ell = r-1 \\ \dfrac{(2a)^m - 2^{m(r+t)}}{2a - 2^{r+t}} & \text{if } a \neq 2^{r+t-1} \end{cases}
\end{aligned}
$$

We have checked that our general formula for $x_n^{(r,t)}(a)$ satisfies this equality when $n = 2^m$ with *Mathematica*.

## Proof of the main result

### Some notations

**1**. Throughout this paper we shall use the following notations related to binary decompositions of natural numbers. For every $n \in \mathbb{N}$, we shall write its binary decomposition as

$$
n = \sum_{j=1}^{s_n} 2^{q_j(n)}, \quad \text{with } 0 \leqslant q_1(n) < \cdots < q_{s_n}(n);
$$

if $n = 0$, we set $s_0 = 0$. With these notations, $s_n$ is the *binary weight* of $n$, that is, the number of 1's in the binary representation of $n$, and, if $n \geqslant 1$, $q_{s_n}(n) = \lfloor \log_2(n) \rfloor$. In order to simplify the notations, we shall set $q_0(n) = 0$.

Notice that, for every $M \in \mathbb{N}_{\geqslant 2}$ and for every natural number $1 \leqslant p < 2^{q_1(M)}$,

$$M + p = \sum_{i=1}^{s_p} 2^{q_i(p)} + \sum_{i=1}^{s_M} 2^{q_i(M)}$$

is the binary decomposition of $M + p$, with $0 \leqslant q_1(p) < \cdots < q_{s_p}(p) < q_1(M) < \cdots < q_{s_M}(M)$, and hence $s_{M+p} = s_p + s_M$ and

$$q_j(M + p) = \begin{cases} q_j(p) & \text{for } j = 1, \ldots, s_p \\ q_{j-s_p}(M) & \text{for } j = s_p + 1, \ldots, s_{M+p} \end{cases} \tag{6}$$

For every $n \in \mathbb{N}$ and $i = 1, \ldots, s_n$, let

$$M_i(n) = \sum_{j=i}^{s_n} 2^{q_j(n)} = 2^{q_i(n)} \left\lfloor \frac{n}{2^{q_i(n)}} \right\rfloor.$$

**2**. For every $n \in \mathbb{N}$, let $\varphi_0(n) = \lfloor n/2 \rfloor$ and $\varphi_1(n) = \lceil n/2 \rceil$ and, for every $m \geqslant 1$ and for every sequence $b_m \ldots b_0 \in \{0, 1\}^{m+1}$, let

$$\varphi_{b_m \ldots b_0}(n) = \varphi_{b_m}(\varphi_{b_{m-1} \ldots b_0}(n)).$$

By Thompson's Rounding Lemma [43], for every sequence $b_m \ldots b_0 \in \{0, 1\}^{m+1}$,

$$\varphi_{b_m \ldots b_0}(n) = \left\lfloor \frac{n + \sum_{i=0}^{m} b_i 2^i}{2^{m+1}} \right\rfloor. \tag{7}$$

**3**. For every $m \in \mathbb{N}$, let $B_m$ denote the $m$-th Bernoulli number of the first kind. Recall that these Bernoulli numbers can be defined, starting with $B_0 = 1$, by means of the recurrence

$$\sum_{k=0}^{m} \binom{m+1}{k} B_k = 0, \quad m \geqslant 1. \tag{8}$$

Let moreover $B_m(x) \in \mathbb{Q}[x]$ be the Bernoulli polynomial of degree $m$, defined by

$$B_m(x) = \sum_{k=0}^{m} \binom{m}{k} B_k x^{m-k}.$$

We list below several well-known properties of the Bernoulli numbers and polynomials that we shall use in our proofs, frequently without any further notice. For these and other properties of the Bernoulli numbers and polynomials, see for instance [44, Ch. 23].

$$B_1 = -1/2 \text{ and } B_{2i+1} = 0 \text{ for every } i \geqslant 1 \tag{9}$$

$$B_m = B_m(0) = (-1)^m B_m(1) \tag{10}$$

$$B_m(x + y) = \sum_{k=0}^{m} \binom{m}{k} B_k(x) y^{m-k} \tag{11}$$

In particular,

$$
\begin{aligned}
B_m(2) \quad &= B_m(1+1) = \sum_{k=0}^{m}\binom{m}{k}B_k(1) \qquad &\text{(by (11))}\\[2mm]
&= \sum_{k=0}^{m}\binom{m}{k}(-1)^k B_k \qquad &\text{(by (10))}\\[2mm]
&= \sum_{k=0}^{m}\binom{m}{k}B_k - 2mB_1 = \sum_{k=0}^{m}\binom{m}{k}B_k + m \qquad &\text{(by (9))}\\[2mm]
&= B_m(1) + m = (-1)^m B_m + m \qquad &\text{(again by (10))}
\end{aligned}
\tag{12}
$$

**4.** For every $d \in \mathbb{N}$, $n \in \mathbb{N}_{\geqslant 1}$, and $x \in \mathbb{R} \setminus \{0\}$, let

$$
T(d,n,x) = \sum_{k=0}^{n-1} k^d x^k
$$

The value of $T(d, n, 1)$ is given by *Faulhaber's formula* [29, (6.78)]: for every $d \in \mathbb{N}$ and $n \in \mathbb{N}_{\geqslant 1}$,

$$
\sum_{k=0}^{n-1} k^d = \frac{1}{d+1}\sum_{j=0}^{d}\binom{d+1}{j}B_j n^{d+1-j}
$$

We shall also use this formula in the following, equivalent way:

$$
\sum_{k=1}^{n-1} k^d = \frac{1}{d+1}\left(\sum_{j=0}^{d+1}\binom{d+1}{j}B_j n^{d+1-j} + (-1)^d B_{d+1}\right)
\tag{13}
$$

The equivalence stems from the fact that if $d \geqslant 1$, $(-1)^d B_{d+1} = -B_{d+1}$, while when $d = 0$, $(-1)^d B_{d+1} = B_1 = -1/2$.

When $x \neq 1$, the double sequence $T$ satisfies the recurrence

$$
T(d,n,x) = \frac{x}{1-x}\sum_{p=0}^{d-1}\binom{d}{p}T(p,n,x) - \frac{x^n}{1-x}n^d, \quad d\geqslant 1.
$$

In particular, when $x \neq 1$,

$$
T(1,n,x) = \frac{x}{1-x}\left(T(0,n,x) - nx^{n-1}\right)
\tag{14}
$$

We shall use these double sequences $T(d, n, x)$ in order to unify some notations and proofs, but, as we have already encountered in the statement of our main result, we actually only need to know closed formulas for them when $d = 0, 1$:

$$
\begin{aligned}
&T(0,n,1) = n, \qquad &&T(0,n,x) = \frac{x^n-1}{x-1} \quad \text{for } x \neq 1\\[3mm]
&T(1,n,1) = \binom{n}{2}, \qquad &&T(1,n,x) = \frac{nx^n(x-1) - x(x^n-1)}{(x-1)^2} \quad \text{for } x \neq 1
\end{aligned}
\tag{15}
$$

## Some technical lemmas

We begin by providing closed formulas for several sums that we shall often encounter in our computations.

**Lemma 4.** *For every* $d \in \mathbb{N}$, $n \in \mathbb{N}_{\geqslant 1}$, *and* $x \in \mathbb{R} \setminus \{0\}$:

(a) $\sum_{k=1}^{n-1} q_{s_k}(k)^d x^{q_{s_k}(k)} = T(d, q_{s_n}(n), 2x) + n q_{s_n}(n)^d x^{q_{s_n}(n)} - q_{s_n}(n)^d (2x)^{q_{s_n}(n)}$

(b) $\sum_{k=0}^{n-1} \sum_{i=1}^{s_k} q_i(k)^d x^{q_i(k)} = \sum_{i=1}^{s_n} 2^{q_i(n)-1} T(d, q_i(n), x) + \sum_{i=1}^{s_n} q_i(n)^d x^{q_i(n)} (n - M_i(n))$

The next lemma is a key step in the obtention of finite explicit expressions for the solutions of the recurrences considered in this paper.

**Lemma 5.** *Let* $(z_{l,p})_{(l,p) \in \mathbb{N}^2}$ *be a double sequence satisfying*:

*(a) For every* $p \in \mathbb{N}$, $z_{0,p} = 0$

*(b) For every* $l, p > 0$,

$$z_{l,p} = 2z_{l-1,p} + \sum_{q=0}^{p-1} \binom{p}{q} 2^{(p-q)(l-1)} z_{l-1,q}$$

*Then, for every* $l \geqslant 0$ *and for every* $p > 0$,

$$z_{l,p} = -\frac{2^l}{p+1} \sum_{j=1}^{l-1} \left( \sum_{i=1}^{p} \binom{p+1}{i} (2^i - 1) 2^{(p-i)l+(i-1)j} B_i \right) z_{j,0}.$$

For every $l, m, p, d \in \mathbb{N}$, with $p < m$, and for every given $a \in \mathbb{R} \setminus \{0\}$, let

$$\gamma_l^{(d,p,m)} = \sum_{k=1}^{2^l-1} \sum_{i=1}^{s_k} q_i(k)^d (2^{-m}a)^{q_i(k)} M_{i+1}(k)^p$$

In the last part of our computations we shall use an alternative expression for $\gamma_l^{(d,p,m)}$ which we derive now as an application of the last lemma.

We have that $\gamma_0^{(d,p,m)} = 0$ by definition, and, by Lemma 4.(b),

$$\gamma_l^{(d,0,m)} = \sum_{k=1}^{2^l-1} \sum_{i=1}^{s_k} q_i(k)^d (a2^{-m})^{q_i(k)} = 2^{l-1} T(d, l, a2^{-m}) \tag{16}$$

The case when $l, p > 0$ is covered by the next lemma.

**Lemma 6.** *For every* $l, m, p, d \in \mathbb{N}$, *with* $p < m$,

$$\gamma_l^{(d,p,m)} = \frac{a^{l-1} 2^{-(m-1)(l-1)+pl}}{p+1} \sum_{t=1}^{l-1} (l-t-1)^d (a^{-1} 2^{m-p-1})^t (B_{p+1}(2^t) - B_{p+1})$$

$$+ (a2^{-(m-1)})^{l-1} (l-1)^d \cdot \delta_{p=0,l>0}$$

*where* $\delta_{p=0,\, l>0} = 1$ *if* $p = 0$ *and* $l > 0$, *and* $\delta_{p=0,\, l>0} = 0$ *otherwise.*

## Statement of the problem

Let $P(x, y) = \sum_{r,t \geqslant 0} b_{r,t} x^r y^t \in \mathbb{R}[x, y]$ be a bivariate polynomial and $a \in \mathbb{R} \setminus \{0\}$, and consider the recurrence equation

$$x_n = a \cdot x_{\lceil n/2 \rceil} + a \cdot x_{\lfloor n/2 \rfloor} + P(\lceil n/2 \rceil, \lfloor n/2 \rfloor), \quad n \geqslant 2. \tag{17}$$

**Lemma 7.** *If $(x_n^0)_n$ is the solution of* (17) *with initial condition $x_1^0 = 0$, then*

$$\bar{x}_n = x_n^0 + ((2a)^{q_{s_n}(n)} + (2a - 1)(n a^{q_{s_n}(n)} - (2a)^{q_{s_n}(n)})) x_1$$

*is its solution with initial condition $x_1$*

*Proof.* Since $\bar{x}_1 = x_1$, we must prove that $(\bar{x}_n)_n$ satisfies (17). Since $(x_n^0)_n$ already satisfies it, it is enough to check that, for every $n \geqslant 2$,

$$(2a)^{q_{s_n}(n)} + (2a - 1)(n a^{q_{s_n}(n)} - (2a)^{q_{s_n}(n)})$$
$$= a((2a)^{q_{s_{\lceil n/2 \rceil}}(\lceil n/2 \rceil)} + (2a - 1)(\lceil n/2 \rceil a^{q_{s_{\lceil n/2 \rceil}}(\lceil n/2 \rceil)} - (2a)^{q_{s_{\lceil n/2 \rceil}}(\lceil n/2 \rceil)}))$$
$$+ a((2a)^{q_{s_{\lfloor n/2 \rfloor}}(\lfloor n/2 \rfloor)} + (2a - 1)(\lfloor n/2 \rfloor a^{q_{s_{\lfloor n/2 \rfloor}}(\lfloor n/2 \rfloor)} - (2a)^{q_{s_{\lfloor n/2 \rfloor}}(\lfloor n/2 \rfloor)}))$$

The simplest way to do it is by distinguishing two cases; to simplify the notations, we denote in the rest of this proof $q_{s_n}(n)$ by $q$.

(i) If $n \neq 2^{q+1} - 1$, then $q_{s_{\lceil n/2 \rceil}}(\lceil n/2 \rceil) = q_{s_{\lfloor n/2 \rfloor}}(\lfloor n/2 \rfloor) = q - 1$ and then

$$a((2a)^{q-1} + (2a - 1)(\lceil n/2 \rceil a^{q-1} - (2a)^{q-1}) + (2a)^{q-1} + (2a - 1)(\lfloor n/2 \rfloor a^{q-1} - (2a)^{q-1}))$$
$$= a(2(2a)^{q-1} + (2a - 1)(n a^{q-1} - 2(2a)^{q-1})) = (2a)^q + (2a - 1)(n a^q - (2a)^q)$$

(ii) If $n = 2^{q+1} - 1$, then $\lceil n/2 \rceil = 2^q$ and $\lfloor n/2 \rfloor = 2^q - 1$, and then

$$a((2a)^q + (2a - 1)(2^q a^q - (2a)^q) + (2a)^{q-1} + (2a - 1)((2^q - 1) a^{q-1} - (2a)^{q-1}))$$
$$= 2^{q+1} a^{q+1} - (2a - 1) a^q = (2a)^q + (2a - 1)((2^{q+1} - 1) a^q - (2a)^q)$$

**Remark 8.** In particular, if $p = 0$, the solution of (17) with initial condition $x_1$ is

$$x_n = ((2a)^{q_{s_n}(n)} + (2a - 1)(n a^{q_{s_n}(n)} - (2a)^{q_{s_n}(n)})) x_1,$$

as it was already proved in [30, Lem. 21].

Therefore, from now on we shall only be concerned with the solution of (17) with initial condition $x_1 = 0$. Now, if, for every $(r, t) \in \mathbb{N}^2$ such that $b_{r,t} \neq 0$, we know the solution $(x_n^{(r,t)})_n$ of the recurrence

$$x_n^{(r,t)} = a \cdot x_{\lceil n/2 \rceil}^{(r,t)} + a \cdot x_{\lfloor n/2 \rfloor}^{(r,t)} + \left\lceil \frac{n}{2} \right\rceil^r \cdot \left\lfloor \frac{n}{2} \right\rfloor^t, \quad n \geqslant 2, \tag{18}$$

with initial condition $x_1^{(r,t)} = 0$, then the solution of (17) with initial condition $x_1 = 0$ will be

$$x_n = \sum_{r,t} b_{r,t} x_n^{(r,t)}.$$

So, we are finally led to consider, for any $r, t \in \mathbb{N}$, the equation (18). Let $x_n^{(r,t)}$ be its solution with $x_1^{(r,t)} = 0$.

## The sequence of differences of consecutive terms $y_n^{(r,t)}$

Let $y_n^{(r,t)} = x_n^{(r,t)} - x_{n-1}^{(r,t)}$, with $y_1^{(r,t)} = 0$, and

$$x_n^{(r,t)} = \sum_{k=1}^{n} y_k^{(r,t)}. \tag{19}$$

Notice that

$$y_2^{(r,t)} = x_2^{(r,t)} - x_1^{(r,t)} = a \cdot x_1^{(r,t)} + a \cdot x_1^{(r,t)} + 1 - x_1^{(r,t)} = 1.$$

Next result gives an expression for $y_n^{(r,t)}$ that will be suitable to our purposes. To simplify the notations, in its statement and henceforth, for every $d, m, n \in \mathbb{N}$, we set

$$S_n^{(d,m)} = \sum_{j=1}^{s_n-1} q_j(n)^d (2^{-m}a)^{q_j(n)} M_{j+1}(n)^m.$$

Moreover, we shall let $\ell = \log_2(a) - t$ when $a > 0$, and we shall use the following Kronecker delta:

$$\delta_\ell = \begin{cases} 1 & \text{if } a > 0, \ r > 0, \ \text{and } \ell \in \{0, \dots, r-1\} \\ 0 & \text{otherwise} \end{cases}$$

**Proposition 9.** *For every $n \geqslant 2$,*

$$y_n^{(r,t)} = a^{q_{s_{n-1}}(n-1)} + \sum_{\substack{l=0 \\ l \neq \ell}}^{r-1} \frac{\binom{r}{l}}{2^{t+l} - a} \left( (n-1)^{t+l} - a^{q_{s_{n-1}}(n-1)} \right)$$

$$+ \sum_{i=0}^{r+t-1} \left( 2^{-i} \binom{r+t}{i} - 2^{-i+1} \binom{r}{i-t} - \sum_{\substack{l=i-t+1 \\ l \neq \ell}}^{r-1} \frac{\binom{r}{l}\binom{t+l}{i}}{2^{t+l} - a} \right) S_{n-1}^{(0,i)}$$

$$+ \delta_\ell \cdot \frac{1}{a} \binom{r}{\ell} \sum_{j=0}^{s_{n-1}-1} M_{j+1}(n-1)^{t+\ell} (q_{j+1}(n-1) - q_j(n-1))$$

**Remark 10.** When $r = t = 0$, the formula in the last proposition simply says $y_n^{(r,t)} = a^{q_{s_{n-1}}(n-1)}$.

In the expression for $y_n^{(r,t)}$ given in the last proposition, the term

$$\delta_\ell \cdot \frac{1}{a} \sum_{j=0}^{s_{n-1}-1} M_{j+1}(n-1)^{t+\ell} (q_{j+1}(n-1) - q_j(n-1))$$

is different from 0 only when $r > 0$ and $a \in \{2^t, \dots, 2^{r+t-1}\}$. When $t + \ell = 0$, that is, when

$a = 1$, the sum in it simplifies to

$$\sum_{j=0}^{s_{n-1}-1} M_{j+1}(n-1)^0 (q_{j+1}(n-1) - q_j(n-1)) = q_{s_{n-1}}(n-1) \tag{20}$$

And notice that this sum contributes in this $a = 1$ case to $y_n^{(r,t)}$ only when $r > 0$ and $a = 1 \in \{2^t, \ldots, 2^{r+t-1}\}$, that is, when $r > 0$ and $t = 0$. When $t + \ell > 0$, and recalling that $2^{t+\ell} = a$, this sum is

$$\sum_{j=0}^{s_{n-1}-1} M_{j+1}(n-1)^{t+\ell} (q_{j+1}(n-1) - q_j(n-1))$$

$$= \sum_{j=1}^{s_{n-1}-1} q_j(n-1)(M_j(n-1)^{t+\ell} - M_{j+1}(n-1)^{t+\ell}) + 2^{(t+\ell)q_{s_{n-1}}(n-1)} q_{s_{n-1}}(n-1)$$

$$= \sum_{j=1}^{s_{n-1}-1} q_j(n-1)((M_{j+1}(n-1) + 2^{q_j(n-1)})^{t+\ell} - M_{j+1}(n-1)^{t+\ell}) + a^{q_{s_{n-1}}(n-1)} q_{s_{n-1}}(n-1) \tag{21}$$

$$= a^{q_{s_{n-1}}(n-1)} q_{s_{n-1}}(n-1) + \sum_{j=1}^{s_{n-1}-1} q_j(n-1) \sum_{i=0}^{t+\ell-1} \binom{t+\ell}{i} M_{j+1}(n-1)^i 2^{q_j(n-1)(t+\ell-i)}$$

$$= a^{q_{s_{n-1}}(n-1)} q_{s_{n-1}}(n-1) + \sum_{i=0}^{t+\ell-1} \binom{t+\ell}{i} \sum_{j=1}^{s_{n-1}-1} q_j(n-1) M_{j+1}(n-1)^i (a2^{-i})^{q_j(n-1)}$$

$$= a^{q_{s_{n-1}}(n-1)} q_{s_{n-1}}(n-1) + \sum_{i=0}^{t+\ell-1} \binom{t+\ell}{i} S_{n-1}^{(1,i)} \tag{22}$$

where, for every $n, i \in \mathbb{N}$,

$$S_n^{(1,i)} = \sum_{j=1}^{s_n-1} q_j(n)(2^{-i}a)^{q_j(n)} M_{j+1}(n)^i.$$

Notice that when $\delta_\ell = 1$ and $a = 1$, Eq (22) becomes Eq (20).

## Proof of the main result, up to computing the $\alpha$'s

For every $d, n, m \in \mathbb{N}$,

$$\alpha_n^{(d,m)} = \sum_{k=1}^{n-1} S_k^{(d,m)} = \sum_{k=1}^{n-1} \sum_{j=1}^{s_k-1} q_j(k)^d (2^{-m}a)^{q_j(k)} M_{j+1}(k)^m$$

By Proposition 9 and Eq (19), we have

$$
\begin{aligned}
x_n^{(r,t)} &= \sum_{k=2}^{n} y_k^{(r,t)} = \sum_{k=1}^{n-1}\Bigg[ a^{q_{s_k}(k)} + \delta_\ell \cdot \frac{1}{a}\binom{r}{\ell}\sum_{j=0}^{s_k-1} M_{j+1}(k)^{t+\ell}(q_{j+1}(k)-q_j(k)) + \sum_{\substack{l=0\\l\neq\ell}}^{r-1}\frac{\binom{r}{l}}{2^{t+l}-a}\cdot k^{t+l} \\
&\quad - \sum_{\substack{l=0\\l\neq\ell}}^{r-1}\frac{\binom{r}{l}}{2^{t+l}-a}\cdot a^{q_{s_k}(k)} + \sum_{i=0}^{r+t-1}\left(2^{-i}\binom{r+t}{i}-2^{-i+1}\binom{r}{i-t}-\sum_{\substack{l=i-t+1\\l\neq\ell}}^{r-1}\frac{\binom{r}{l}\binom{t+l}{i}}{2^{t+l}-a}\right)S_k^{(0,i)}\Bigg] \\[2mm]
&= \delta_\ell\cdot\frac{1}{a}\binom{r}{\ell}\sum_{k=1}^{n-1}\sum_{j=0}^{s_k-1}M_{j+1}(k)^{t+\ell}(q_{j+1}(k)-q_j(k)) + \left(1-\sum_{\substack{l=0\\l\neq\ell}}^{r-1}\frac{\binom{r}{l}}{2^{t+l}-a}\right)\left(\sum_{k=1}^{n-1}a^{q_{s_k}(k)}\right) \\[2mm]
&\quad + \sum_{\substack{l=0\\l\neq\ell}}^{r-1}\frac{\binom{r}{l}}{2^{t+l}-a}\sum_{k=1}^{n-1}k^{t+l} + \sum_{i=0}^{r+t-1}\left(2^{-i}\binom{r+t}{i}-2^{-i+1}\binom{r}{i-t}-\sum_{\substack{l=i-t+1\\l\neq\ell}}^{r-1}\frac{\binom{r}{l}\binom{t+l}{i}}{2^{t+l}-a}\right)\sum_{k=1}^{n-1}S_k^{(0,i)} \\[2mm]
&= \delta_\ell\cdot\frac{1}{a}\binom{r}{\ell}\sum_{k=1}^{n-1}\sum_{j=0}^{s_k-1}M_{j+1}(k)^{t+\ell}(q_{j+1}(k)-q_j(k)) + \left(1-\sum_{\substack{l=0\\l\neq\ell}}^{r-1}\frac{\binom{r}{l}}{2^{t+l}-a}\right)\left(\sum_{k=1}^{n-1}a^{q_{s_k}(k)}\right) \\[2mm]
&\quad + \sum_{\substack{l=0\\l\neq\ell}}^{r-1}\frac{\binom{r}{l}}{(t+l+1)(2^{t+l}-a)}\left(\sum_{j=0}^{t+l+1}\binom{t+l+1}{j}B_j n^{t+l+1-j}+(-1)^{t+l}B_{t+l+1}\right) \\[2mm]
&\quad + \sum_{i=0}^{r+t-1}\left(2^{-i}\binom{r+t}{i}-2^{-i+1}\binom{r}{i-t}-\sum_{\substack{l=i-t+1\\l\neq\ell}}^{r-1}\frac{\binom{r}{l}\binom{t+l}{i}}{2^{t+l}-a}\right)\alpha_n^{(0,i)}
\end{aligned}
$$

(by Faulhaber's Formula (13))

$$
\begin{aligned}
&= \delta_\ell\cdot\frac{1}{a}\binom{r}{\ell}\sum_{k=1}^{n-1}\sum_{j=0}^{s_k-1}M_{j+1}(k)^{t+\ell}(q_{j+1}(k)-q_j(k)) + \left(1-\sum_{\substack{l=0\\l\neq\ell}}^{r-1}\frac{\binom{r}{l}}{2^{t+l}-a}\right)\left(\sum_{k=1}^{n-1}a^{q_{s_k}(k)}\right) \\[2mm]
&\quad + \sum_{k=1}^{r+t}\left(\sum_{\substack{i=k\\i\neq t+\ell+1}}^{r+t}\frac{\binom{r}{i-t-1}\binom{i}{k}B_{i-k}}{i(2^{i-1}-a)}\right)n^k + \sum_{\substack{i=t+1\\i\neq t+\ell+1}}^{r+t}\frac{\binom{r}{i-t-1}}{i(2^{i-1}-a)}(B_i-(-1)^iB_i) \quad (23) \\[2mm]
&\quad + \sum_{i=0}^{r+t-1}\left(2^{-i}\binom{r+t}{i}-2^{-i+1}\binom{r}{i-t}-\sum_{\substack{l=i-t+1\\l\neq\ell}}^{r-1}\frac{\binom{r}{l}\binom{t+l}{i}}{2^{t+l}-a}\right)\alpha_n^{(0,i)}
\end{aligned}
$$

because

$$\sum_{\substack{l=0 \\ l \neq \ell}}^{r-1} \frac{\binom{r}{l}}{(t+l+1)(2^{t+l}-a)} \left( \sum_{j=0}^{t+l+1} \binom{t+l+1}{j} B_j n^{t+l+1-j} + (-1)^{t+l} B_{t+l+1} \right)$$

$$= \sum_{\substack{l=0 \\ l \neq \ell}}^{r-1} \frac{\binom{r}{l}}{(t+l+1)(2^{t+l}-a)} \left( \sum_{k=0}^{t+l+1} \binom{t+l+1}{k} B_{t+l+1-k} n^k + (-1)^{t+l} B_{t+l+1} \right)$$

$$= \sum_{\substack{i=t+1 \\ i \neq t+\ell+1}}^{r+t} \frac{\binom{r}{i-t-1}}{i(2^{i-1}-a)} \left( \sum_{k=0}^{i} \binom{i}{k} B_{i-k} n^k - (-1)^i B_i \right)$$

$$= \sum_{k=0}^{r+t} \left( \sum_{\substack{i=k \\ i \neq t+\ell+1}}^{r+t} \frac{\binom{r}{i-t-1}\binom{i}{k} B_{i-k}}{i(2^{i-1}-a)} \right) n^k - \sum_{\substack{i=t+1 \\ i \neq t+\ell+1}}^{r+t} \frac{\binom{r}{i-t-1}}{i(2^{i-1}-a)} (-1)^i B_i$$

$$= \sum_{\substack{i=t+1 \\ i \neq t+\ell+1}}^{r+t} \frac{\binom{r}{i-t-1}}{i(2^{i-1}-a)} B_i + \sum_{k=1}^{r+t} \left( \sum_{\substack{i=k \\ i \neq t+\ell+1}}^{r+t} \frac{\binom{r}{i-t-1}\binom{i}{k} B_{i-k}}{i(2^{i-1}-a)} \right) n^k - \sum_{\substack{i=t+1 \\ i \neq t+\ell+1}}^{r+t} \frac{\binom{r}{i-t-1}}{i(2^{i-1}-a)} (-1)^i B_i$$

$$= \sum_{k=1}^{r+t} \left( \sum_{\substack{i=k \\ i \neq t+\ell+1}}^{r+t} \frac{\binom{r}{i-t-1}\binom{i}{k} B_{i-k}}{i(2^{i-1}-a)} \right) n^k + \sum_{\substack{i=t+1 \\ i \neq t+\ell+1}}^{r+t} \frac{\binom{r}{i-t-1}}{i(2^{i-1}-a)} (B_i - (-1)^i B_i)$$

Now, in Eq (23) we have that, by Lemma 4,

$$\sum_{k=1}^{n-1} a^{q_{s_k}(k)} = T(0, q_{s_n}(n), 2a) + n a^{q_{s_n}(n)} - (2a)^{q_{s_n}(n)}$$

$$= \begin{cases} q_{s_n}(n) + n \cdot 2^{-q_{s_n}(n)} - 1 & \text{if } a = 1/2 \\[2ex] \dfrac{(2a)^{q_{s_n}(n)} - 1}{2a - 1} + n \cdot a^{q_{s_n}(n)} - (2a)^{q_{s_n}(n)} & \text{if } a \neq 1/2 \end{cases}$$

and, by (9),

$$\sum_{\substack{i=t+1 \\ i \neq t+\ell+1}}^{r+t} \frac{\binom{r}{i-t-1}}{i(2^{i-1}-a)} (B_i - (-1)^i B_i) = \begin{cases} 1/(a-1) & \text{if } r > 0, \ t = 0, \ a \neq 1 \\ 0 & \text{otherwise} \end{cases}$$

Concerning the first term of Eq (23), which only contributes to $x_n^{(r,t)}$ when $r > 0$ and $a = 2^{t+\ell}$ with $\ell \in \{0, \ldots, r-1\}$, by (22) and Lemma 4 this sum can be expressed as

$$\sum_{k=1}^{n-1}\sum_{j=0}^{s_k-1} M_{j+1}(k)^{t+\ell}(q_{j+1}(k) - q_j(k)) = \sum_{k=1}^{n-1}\left( a^{q_{s_k}(k)}q_{s_k}(k) + \sum_{i=0}^{t+\ell-1}\binom{t+\ell}{i}S_k^{(1,i)}\right)$$

$$= \sum_{k=1}^{n-1} a^{q_{s_k}(k)}q_{s_k}(k) + \sum_{i=0}^{t+\ell-1}\binom{t+\ell}{i}\sum_{k=1}^{n-1}S_k^{(1,i)}$$

$$= T(1, q_{s_n}(n), 2a) + nq_{s_n}(n)a^{q_{s_n}(n)} - q_{s_n}(n)(2a)^{q_{s_n}(n)} + \sum_{i=0}^{t+\ell-1}\binom{t+\ell}{i}\alpha_n^{(1,i)}$$

In summary, this proves the following theorem.

**Theorem 11.** *For every $n \geqslant 2$,*

$$x_n^{(r,t)} = \left(1 - \sum_{\substack{l=0 \\ l \neq \ell}}^{r-1}\frac{\binom{r}{l}}{2^{t+l} - a}\right)\left(T(0, q_{s_n}(n), 2a) + na^{q_{s_n}(n)} - (2a)^{q_{s_n}(n)}\right)$$

$$+ \sum_{k=1}^{r+t}\left(\sum_{\substack{i=k \\ i \neq t+\ell+1}}^{r+t}\frac{\binom{r}{i-t-1}\binom{i}{k}B_{i-k}}{i(2^{i-1} - a)}\right)n^k + \frac{1}{a-1}\cdot\delta_{r>0,t=0,a\neq 1}$$

$$+ \sum_{i=0}^{r+t-1}\left(2^{-i}\binom{r+t}{i} - 2^{-i+1}\binom{r}{i-t} - \sum_{\substack{l=i-t+1 \\ l \neq \ell}}^{r-1}\frac{\binom{r}{l}\binom{t+l}{i}}{2^{t+l} - a}\right)\alpha_n^{(0,i)}$$

$$+ \delta_\ell\cdot\frac{1}{a}\binom{r}{\ell}\left(T(1, q_{s_n}(n), 2a) + nq_{s_n}(n)a^{q_{s_n}(n)} - q_{s_n}(n)(2a)^{q_{s_n}(n)} + \sum_{i=0}^{t+\ell-1}\binom{t+\ell}{i}\alpha_n^{(1,i)}\right)$$

*where $\delta_{r>0, t=0, a\neq 1} = 1$ if $r > 0$, $t = 0$, and $a \neq 1$, and it is 0 otherwise.*

It remains to compute the terms

$$\alpha_n^{(d,i)} = \sum_{k=1}^{n-1}\sum_{j=1}^{s_k-1} q_j(k)^d (2^{-i}a)^{q_j(k)}M_{j+1}(k)^i, \qquad d = 0, 1.$$

And notice that we only need to compute the $\alpha^{(1,i)}$'s when $r \geqslant 1$ and $a = 2^{t+\ell}$, for some $\ell \in \{0, \ldots, r-1\}$ with $t + \ell > 0$. In particular, in the computation of $\alpha^{(1,i)}$ we can assume that $a \neq 1, 1/2$, in order to avoid discussing cases that are unrelevant for our main result.

## Computation of the $\alpha$'s

Recall that we are interested in closed formulas for $\alpha_n^{(d,m)}$ when $d = 0$ as well as when $d = 1$ and specific values of $a$ which in particular exclude the cases $a = 1, 1/2$. Anyway, it is easier to prove a general expression for them.

**Proposition 12**. *For every $n \geqslant 1$*:

$$
\alpha_n^{(d,m)} = \frac{1}{2(m+1)} \sum_{i=1}^{s_n} \sum_{j=0}^{m} \binom{m+1}{j} B_j 2^j M_i^{m+1-j} (T(d, q_i, a2^{j-m}) - T(d, q_{i-1}, a2^{j-m}))
$$

$$
+ \sum_{i=1}^{s_n-1} q_i^d (a2^{-m})^{q_i} (n - M_i) M_{i+1}^m - T(d, q_{s_n}(n), 2a) \cdot \delta_{m=0}
$$

with $\delta_{m=0} = 1$ if $m = 0$ and $\delta_{m=0} = 0$ if $m > 0$.

**Remark 13**. Notice that the last proposition implies that

$$
\alpha_n^{(0,0)} =
\begin{cases}
\displaystyle \sum_{j=1}^{s_n} 2^{q_j(n)-1}(q_j(n) - 2M_j(n)) + (s_n - 1)n + 1 & \text{if } a = 1 \\[3ex]
\displaystyle \sum_{j=1}^{s_n-1} 2^{-q_j(n)}(n - M_j(n)) + n - q_{s_n} - s_n & \text{if } a = 1/2 \\[3ex]
\displaystyle \sum_{j=1}^{s_n-1} a^{q_j(n)}(n - M_j(n)) + \frac{1}{2(a-1)}\left(\sum_{j=1}^{s_n}(2a)^{q_j(n)} - n\right) - \frac{(2a)^{q_{s_n}(n)} - 1}{2a - 1} & \text{if } a \neq 1, 1/2
\end{cases}
$$

and, when $a \neq 1, 1/2$,

$$
\alpha_n^{(1,0)} = \sum_{i=1}^{s_n} 2^{q_i(n)-1} T(1, q_i(n), a) + \sum_{i=1}^{s_n-1} q_i(n) a^{q_i(n)}(n - M_i(n)) - T(1, q_{s_n}(n), 2a)
$$

$$
= \sum_{i=1}^{s_n-1} q_i(n) a^{q_i(n)}(n - M_i(n)) + \frac{1}{2(a-1)} \sum_{i=1}^{s_n}(2a)^{q_i(n)} q_i(n) - \frac{a}{2(a-1)^2} \sum_{i=1}^{s_n}(2a)^{q_i(n)}
$$

$$
- \frac{1}{2a-1}(2a)^{q_{s_n}(n)} q_{s_n}(n) + \frac{2a}{(2a-1)^2}(2a)^{q_{s_n}(n)} + \frac{a}{2(a-1)^2} n - \frac{2a}{(2a-1)^2}
$$

## Discussion and conclusions

Divide-and-conquer dividing by a half recurrences like (1) appear in many areas of applied mathematics, and they are usually "solved" by computing a bound for their growing order with the aid of a Master Theorem. This is usually enough in many applications, like for instance the asymptotic analysis of algorithms, but in other applications it may be necessary, or at least satisfactory, to obtain an explicit solution of the recurrence: for instance, we have mentioned in the introduction the need for such explicit solutions in the context of the normalization of phylogenetic balance indices. In a previous attempt to solve this problem, Hwang, Janson and Tsai [31] gave an explicit solution of Eq (1) when $a = 1$ and under very general conditions on the independent term, or *toll function*, $p(n)$, in terms of sums of infinite series. But then, if one does not know how to sum the series, the explicit solution slips away.

In this paper we have restricted ourselves to the case when the toll function is a polynomial in $\lfloor n/2 \rfloor$ and $\lceil n/2 \rceil$. In this case, and for an arbitrary coefficient $a$, we are able to give a finite explicit expression for the solution of (1) in terms only of the binary expansion of $n$. The existence of such a closed formula is not surprising, but, for instance, it cannot be deduced directly from Hwang, Janson and Tsai solution. An implementation of our solution in Python is available at https://github.com/biocom-uib/divide_and_conquer.

The restriction to a polynomial independent term is a serious limitation for the usefulness of our formula. It remains as future work to use the same approach to obtain explicit solutions

for other specific types of toll functions, including for instance terms affected by coefficients $(-1)^{\lfloor n/2 \rfloor}$ and $(-1)^{\lceil n/2 \rceil}$ (we have seen in Example 4 how our approach allows to deal with terms affected by coefficients $(-1)^n$) or rational functions in $\lfloor n/2 \rfloor$ and $\lceil n/2 \rceil$. The main obstacle in the derivation of such formulas is the generalization to their setting of the key Lemma 5, used to simplify the sums $\gamma_l^{(d,p,m)}$ in Lemma 6. Also it would be interesting to obtain similar explicit solutions when the coefficients of $x_{\lfloor n/2 \rfloor}$ and $x_{\lceil n/2 \rceil}$ are different, and more specifically when one of them is 0. This is done for homogeneous equations in [45, Lem. 4], but the expression provided therein for the solution $x_n$ involves sums from 1 to $n$, and not up to $\lfloor \log_2(n) \rfloor$ as those obtained with our techniques.

## Supporting information

**S1 File. Proofs of several results.** The file provides the detailed proofs of Lemmas 4–6 and Propositions 9 and 12.
(PDF)

## Acknowledgments

We thank G. Valiente for some useful suggestions.

## Author Contributions

**Conceptualization:** Tomás M. Coronado, Arnau Mir, Francesc Rosselló.

**Formal analysis:** Tomás M. Coronado, Arnau Mir, Francesc Rosselló.

**Funding acquisition:** Francesc Rosselló.

**Investigation:** Tomás M. Coronado, Arnau Mir, Francesc Rosselló.

**Methodology:** Tomás M. Coronado, Arnau Mir, Francesc Rosselló.

**Software:** Tomás M. Coronado, Arnau Mir.

**Supervision:** Francesc Rosselló.

**Writing – original draft:** Francesc Rosselló.

**Writing – review & editing:** Tomás M. Coronado, Arnau Mir, Francesc Rosselló.

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
