## [Decision Letter · Decision Letter 0]

22 Jun 2022

PONE-D-22-15094Explicit solution of divide-and-conquer dividing by a half recurrences with polynomial independent termPLOS ONE

Dear Dr. Rosselló,

Thank you for submitting your manuscript to PLOS ONE. After careful consideration, we feel that it has merit but does not fully meet PLOS ONE’s publication criteria as it currently stands. Therefore, we invite you to submit a revised version of the manuscript that addresses the points raised during the review process.

We look forward to receiving your revised manuscript.

Kind regards,

Ashwani Kumar, Ph.D.

Academic Editor

PLOS ONE

Journal Requirements:

"This research was partially supported by the Spanish Ministry of Science, Innovation and

Universities and the European Regional Development Fund through projects PGC2018-096956-B-C43 and PID2021-126114NB-C44 (FEDER/MICINN/AEI)."

"All authors were partially funded by the Spanish Ministry of Science and Innovation and the European Regional Development Fund through projects PGC2018-096956-B-C43 and PID2021-126114NB-C44 (FEDER/MICINN/AEI). The funders had no role in study design, data collection and analysis, decision to publish, or preparation of the manuscript."

Reviewers' comments:

Reviewer's Responses to Questions

**Comments to the Author**

1. Is the manuscript technically sound, and do the data support the conclusions?

Reviewer #1: Yes

Reviewer #2: Yes

2. Has the statistical analysis been performed appropriately and rigorously? 

Reviewer #1: N/A

Reviewer #2: Yes

3. Have the authors made all data underlying the findings in their manuscript fully available?

Reviewer #1: Yes

Reviewer #2: Yes

4. Is the manuscript presented in an intelligible fashion and written in standard English?

Reviewer #1: Yes

Reviewer #2: Yes

5. Review Comments to the Author

Reviewer #1: In this study, the authors explore the explicit solution of divide-and-conquer dividing by a half recurrences with polynomial independent term. The aim and objective of the study are clear to me and work is very interesting. The publication is recommended after addressing the following comments:

1. The abstract should contain answers to the following questions: What problem was ‎studied and why is it important? Add a few applications of the current elaborated problem, and avoid citation in the abstract. Don’t used equation in the abstract, remove it if feasible.

2. The motivation and gap of study is not really highlighted in the last paragraph of the introduction section. The paragraph should elaborate more on the importance, and address/highlight the research contributions.

3. The originality of the paper needs to be stated clearly. It is of important to have sufficient results to justify the novelty of a high-quality journal paper. The Introduction should make a compelling case for why the study is useful along with a clear statement of its novelty or originality by providing relevant information and providing answers to basic questions such as: What is already known in the open literature? What is missing (i.e., research gaps)? What needs to be done, why and how? Clear statements of the novelty of the work should also appear briefly in the Abstract and Conclusions sections.

4. Check all the equations for possible typo mistakes.

5. Must write real application of elaborated problem.

6. Write future direction of the current study.

Reviewer #2: The authors presented a mathematical analysis using divide-and-conquer dividing by a half recurrences. This methodology can be used in many areas of applied mathematics. The mathematical aspects of the paper are interesting; so, I recommend the paper for publication after some modifications.

1. What is new in the model equations and why these are considered?

2. Is it necessary to restrict yourselves to the case when the toll function is a polynomial?

3. When will explicit solution zero?

4. Correlate your work with the earlier ones and provide distinction of your work.

please address above issues in the revised paper.

6. PLOS authors have the option to publish the peer review history of their article (what does this mean?). If published, this will include your full peer review and any attached files.

Reviewer #1: No

Reviewer #2: **Yes: **Sohail Ahmad

---

## [Author Response · Author response to Decision Letter 0]

17 Aug 2022

See the "Response to Reviewers" document

---

## [Decision Letter · Decision Letter 1]

30 Aug 2022

Explicit solution of divide-and-conquer dividing by a half recurrences with polynomial independent term

PONE-D-22-15094R1

Dear Dr. Rosselló,

We’re pleased to inform you that your manuscript has been judged scientifically suitable for publication and will be formally accepted for publication once it meets all outstanding technical requirements.

Kind regards,

Ashwani Kumar, Ph.D.

Academic Editor

PLOS ONE

Additional Editor Comments (optional):

Reviewers' comments:

Reviewer's Responses to Questions

**Comments to the Author**

1. If the authors have adequately addressed your comments raised in a previous round of review and you feel that this manuscript is now acceptable for publication, you may indicate that here to bypass the “Comments to the Author” section, enter your conflict of interest statement in the “Confidential to Editor” section, and submit your "Accept" recommendation.

Reviewer #1: All comments have been addressed

Reviewer #2: All comments have been addressed

2. Is the manuscript technically sound, and do the data support the conclusions?

Reviewer #1: Yes

Reviewer #2: Yes

3. Has the statistical analysis been performed appropriately and rigorously? 

Reviewer #1: N/A

Reviewer #2: Yes

4. Have the authors made all data underlying the findings in their manuscript fully available?

Reviewer #1: Yes

Reviewer #2: Yes

5. Is the manuscript presented in an intelligible fashion and written in standard English?

Reviewer #1: Yes

Reviewer #2: Yes

6. Review Comments to the Author

Reviewer #1: (No Response)

Reviewer #2: I appreciate the effort of authors. The manuscript has been well revised. I recommend the paper for publication.

7. PLOS authors have the option to publish the peer review history of their article (what does this mean?). If published, this will include your full peer review and any attached files.

Reviewer #1: **Yes: **Bagh Ali

Reviewer #2: **Yes: **Sohail Ahmad

---

## [Editor Report · Acceptance letter]

2 Sep 2022

PONE-D-22-15094R1 

Explicit solution of divide-and-conquer dividing by a half recurrences with polynomial independent term 

Dear Dr. Rosselló:

I'm pleased to inform you that your manuscript has been deemed suitable for publication in PLOS ONE. Congratulations! Your manuscript is now with our production department. 

Kind regards, 

on behalf of

Dr. Ashwani Kumar 

Academic Editor

PLOS ONE